# AdaStop: adaptive statistical testing for sound comparisons of Deep RL agents

**Timothée Mathieu**                                       *timothee.mathieu@inria.fr*
*Inria, Université de Lille, CNRS, Centrale Lille, UMR 9189 – CRIStAL*

**Riccardo Della Vecchia**                              *ric.della.vecchia@gmail.com*
*Inria, Université de Lille, CNRS, Centrale Lille, UMR 9189 – CRIStAL*

**Alena Shilova**                                              *alena.shilova@inria.fr*
*Inria, Université de Lille, CNRS, Centrale Lille, UMR 9189 – CRIStAL*

**Matheus Medeiros Centa**                      *matheus.medeiros-centa@inria.fr*
*Université de Lille, Inria, CNRS, Centrale Lille, UMR 9189 – CRIStAL*

**Hector Kohler**                                            *hector.kohler@inria.fr*
*Université de Lille, Inria, CNRS, Centrale Lille, UMR 9189 – CRIStAL*

**Odalric-Ambrym Maillard**                          *odalric.maillard@inria.fr*
*Inria, Université de Lille, CNRS, Centrale Lille, UMR 9189 – CRIStAL*

**Philippe Preux**                                          *philippe.preux@inria.fr*
*Université de Lille, Inria, CNRS, Centrale Lille, UMR 9189 – CRIStAL*

**Reviewed on OpenReview:** *https://openreview.net/forum?id=lXyZr9TLEU*

## Abstract

Recently, the scientific community has questioned the statistical reproducibility of many empirical results, especially in the field of machine learning. To contribute to the resolution of this reproducibility crisis, we propose a theoretically sound methodology for comparing the performance of a set of algorithms. We exemplify our methodology in Deep Reinforcement Learning (Deep RL). The performance of one execution of a Deep RL algorithm is a random variable. Therefore, several independent executions are needed to evaluate its performance. When comparing algorithms with random performance, a major question concerns the number of executions to perform to ensure that the result of the comparison is theoretically sound. Researchers in Deep RL often use less than 5 independent executions to compare algorithms: we claim that this is not enough in general. Moreover, when comparing more than 2 algorithms at once, we have to use a multiple tests procedure to preserve low error guarantees. We introduce AdaStop, a new statistical test based on multiple group sequential tests. When used to compare algorithms, AdaStop adapts the number of executions to stop as early as possible while ensuring that enough information has been collected to distinguish algorithms that have different score distributions. We prove theoretically that AdaStop has a low probability of making a (family-wise) error. We illustrate the effectiveness of AdaStop in various use-cases, including toy examples and Deep RL algorithms on challenging Mujoco environments. AdaStop is the first statistical test fitted to this sort of comparisons: it is both a significant contribution to statistics, and an important contribution to computational studies performed in reinforcement learning and in other domains.

# 1 Introduction

In many fields of computer science, it is customary to perform an experimental investigation to compare the practical performance of two or more algorithms. When the behavior of an algorithm is non-deterministic (for instance because the algorithm is non-deterministic, or because the data it is fed upon is random), the performance of the algorithm is a random variable. Then, the way to do this comparison is not clear. Usually, one executes the algorithm (or rather an implementation of the algorithm: this point will soon be clarified) several times in order to obtain an average performance and its variability. How much "several" is depends on the authors, and some contingencies: thanks to the law of large numbers, one may think that the larger the better, the more accurate the estimates of the average and the variability. This may be a satisfactory answer, but when a single execution lasts days or even weeks or months (such as LLM training), performing such a "large enough" number of executions is impossible.

To illustrate this point, let us consider the field of deep reinforcement learning (Deep RL), that is reinforcement learning algorithms that use a neural network to represent what they learn. We surveyed all deep RL papers published in the proceedings of the International Conference on Machine Learning in 2022 (see Fig.1a). In the vast majority of these papers, only a few executions have been performed: among the 18 papers using the Mujoco tasks, only 3 papers performed more than 10 runs, and 11 papers used only 3 to 5 runs. This begs the question: if we can be confident that 3 executions are not enough, how many executions would be enough to draw statistically significant conclusions? Conversely, are 80 executions not too much? This is not only a matter of computation time and computational resources occupation. Each execution contributes to pollute our planet and to the climate change. There is no answer to these questions today. Moreover, if someone redoes the comparison of the same algorithms on the same task, the conclusions should be the same: this concept is known as *statistical reproducibility* (Agarwal et al., 2021; Colas et al., 2019; Goodman et al., 2016) and should be a feature of any good experimental design.

This paper provides a partial answer to the problem: we introduce AdaStop, a new statistical test tailored to the problem of small sample sizes in which we cannot suppose that the data are Gaussian. On the application side, we demonstrate the use of AdaStop in practice and show its efficiency in dealing with comparison of Deep RL algorithms. On the theory side, we show the usual non-asymptotic and asymptotic guarantees of such a nonparametric test: we show that for large sample sizes, AdaStop controls the false positive rate appropriately, and we prove a non-asymptotic bound on false positive (*i.e.* family-wise error) when comparing the distributions. We keep as future work the question of the non-asymptotic comparison of the means of distributions.

This paper is accompanied by a software program that implements the test which is very easy to use. In short, in this paper, we provide a methodological approach and its actual implementation to compare the performance of algorithms having random performance in a statistically significant way, while trying to minimize the computational effort to reach such a conclusion. Such experimental investigations arise in various fields of computational AI such as machine learning, and computational optimization. We will use AdaStop in the field of reinforcement learning to illustrate this paper, but its application to other fields of computational AI is straightforward, as well as in many other fields of science using computational studies to compare the performance of algorithms.

The organization of the paper is as follows: in an informal way, we detail the requirements that has to fulfil a statistical test to fit our expectations in Section 2. In this section, we also make clear the limitations of AdaStop. Section 3 introduces the main ingredients of our test, AdaStop. Section 4 in the formal analysis of the properties of AdaStop. Proofs are established in appendices C to F. We illustrate the use of AdaStop in Section 5 before concluding. Appendix A lists all the notations used in this paper. Appendix B provides a minimal exposition of the main concepts of hypothesis testing for readers who are not trained in statistics.

To reproduce the experiments of this paper, the python code is freely available on GitHub at `https://github.com/TimotheeMathieu/Adaptive_stopping_MC_RL`. In addition, we provide a library and command-line tool that can be used independently: the AdaStop Python package is available at `https://github.com/TimotheeMathieu/adastop`.

## 2 Statistical reproducibility in RL through the lens of statistical tests

In this section we begin by identifying the key concepts necessary to discuss reproducibility in the case of Deep RL. Then we investigate the pros and cons of statistical tests to answer the statistical reproducibility problem in Deep RL, and we compare this methodology to current practices in experimental Deep RL.

### 2.1 Definitions

First, we define some key terms that we use in the rest of the paper.

As raised above, we do not compare algorithms but a certain implementation of an algorithm using a certain set of values for its hyperparameters. In D. Knuth's spirit (Knuth, 1968), we use the term **algorithm** in its usual meaning in computer science as the description of the basic operations required to transform a certain input into a certain output. By basic operations we mean the use of variables and simple operations (arithmetical, logical, etc), along with assignments to variables, sequences of instructions, tests and loops. Such an algorithm typically has some hyperparameters that control its behavior (a threshold, the dimension of the input domain, how a certain variable decays in time, the architecture of a neural network, etc). In this regard, we may say that as presented in (Schulman et al., 2017), PPO is an algorithm. However, one should be cautious that many aspects of PPO may be defined in various ways, and that the notion of "the PPO algorithm" is not as clearly defined as *e.g.* "the quicksort algorithm". The same may be said for all Deep RL algorithms for which there exist many variations.

Let us make clear the distinction between parameters and hyperparameters: parameters are learned from data, while hyperparameters are set a priori. For instance, the weights of a neural network are parameters, while the architecture of the network is a hyperparameter (unless this architecture is also learned during the training of the agent, which is far from being a common practice in Deep RL).

We use the term **agent** to refer to a certain implementation of an algorithm along with the value of its hyperparameters. If an agent is run several times, the value of the hyperparameters is the same at the beginning of each run. A special case concerns the seed of the pseudo-random number generator. Deep RL algorithms typically use a different seed to initialize each run. So the seed is not part of the definition of an agent. In the context of reinforcement learning, we call **policy** a trained agent. A policy is a decision function that maps observations to actions.

The **score** of an agent is a numerical value that quantifies its performance. Its definition is really up to the experimenter and the objective of the experiments she designed. In the case of an RL agent, the score is usually a numerical value computed from policy evaluations. For instance, the score used in a given experiment may be the mean episodic return, or its variance, or many other quantities (running time, memory consumption, number of updates, etc). In most Deep RL research, agents are compared as follows: 1) an agent is trained on one random seed, 2) after training, the policy is evaluated for a given number of episodes to compute one score, 3) this evaluation is repeated $K$ times eventually providing $K$ scores, 4) then a statistic is computed over these $K$ scores, 5) some conclusion about the performance of the various agents/algorithms is drawn. We suppose that enough policy evaluations have been performed to account for the possible stochasticity of both the policy and the environment. ADASTOP is concerned with having a significant comparison of the theoretical mean of the scores while using as few random seeds as possible.

### 2.2 Ingredients for an appropriate statistical test

Our goal is to provide a statistical test to decide whether one agent performs better than some others. However, strictly speaking, we really decide whether two agents perform equally or not by comparing the statistic of their scores measured on a set of (evaluation) runs, up to some confidence $\alpha$. Let us assume that the statistic is the mean. Then, when ADASTOP concludes that the means of the 2 agents differ, it is common practice to rank the agents based on their comparison. However, strictly speaking, this is an abuse and this conclusion is only valid up to some probability which is usually considered so large that its alternative can be rejected. To stress this point, we will write that an agent is "most likely performing better" than an other agent when the test of their mean performances is rejected. We will denote this with the acronym MLB.

The reader should note that this subtlety is not only true for ADASTOP but it is true for all statistical tests based on the equality of the means of two distributions. This point is rarely made clear in publications.

Now, let us list the requirements and difficulties we have to face in the design of such a statistical test.

First, many tests in statistics are defined for Gaussian random variables. However, one quickly figures out that the observed performance of an agent is usually not Gaussian. Fig.1b illustrates this point: we represent the distribution of the performance of 4 agents implementing 4 different RL algorithms (PPO, SAC, DDPG, TRPO): the performance is usually multi-modal, and it is not even a mixture of a few Gaussian distributions as it may seem. This leads us to a nonparametric test. Second, we would like to perform the minimal number of runs. This leads us to the use of a sequential adaptive test that tells us if we need to run the agents a few more times, or if we can take a decision in a statistically significant way with the already collected data. Third, we want to be able to compare more than 2 agents which leads us to multiple testing. Fourth, we want the conclusions of the test to be statistically reproducible, that is, if someone reproduces the execution of the agents and applies the test in the same way as someone else, the conclusion is the same. Fifth, we want to keep the number of runs within reasonable limits: for that purpose, we set a maximum number of scores to collect: if no decision can be made using this budget, the test can not decide whether an agent is MLB than the others. The first 3 requirements call for a nonparametric, sequential, multiple test. A candidate statistical test that may verify all these properties can be found in group sequential permutation test (see the textbook (Jennison & Turnbull, 1999) on general group sequential tests). We use these 5 ingredients to construct ADASTOP.

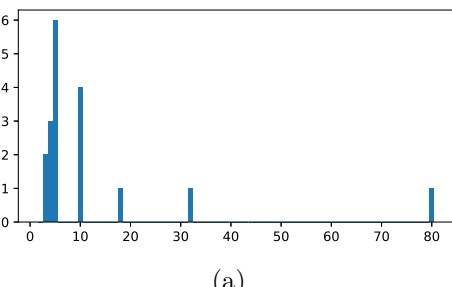 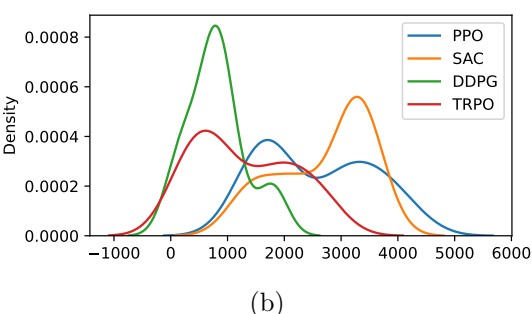

(a)                                             (b)

Figure 1: Motivations for ADASTOP. (a): a census of the number of scores ($N_{score}$) used in RL papers using Mujoco environments published in the proceedings of ICML 2022. (b): estimations of the score distributions of 4 well-known Deep RL agents on Hopper, a Mujoco environment, with $N_{score} = 30$. We see from the census that most experiments used 5 or less scores ($N_{score} \leq 5$) to draw conclusions. Those conclusions are most likely statistically wrong as they are equivalent to drawing 5 samples from distributions similar to the right plot to draw conclusions about the empirical means.

ADASTOP, our proposed statistical test, meets all these expectations. Before diving into the technical details, let us briefly explain how ADASTOP is used in practice. Fig. 2 illustrates an execution of ADASTOP on a small example. Let us suppose that we want to compare the performance of 2 agents, a green agent and a blue agent. Fig. 2 illustrates the sequential nature of the test from top to bottom. Initially, each agent is executed $N = 5$ times. This yields 5 scores for each agent: these 10 initial scores are shown on the top-leftmost part of Fig. 2 labelled `Interim 1, N = 5`, along with their barplots. After the test statistics are computed, ADASTOP decides that this information is not enough to conclude, and more scores are needed. This set of actions (collection of 5 scores for each agent, computation of the statistics, and decision) is known as an **interim**. As no conclusion can be drawn, a second interim is performed: both agents are run 5 more times, yielding 10 more scores. In the middle of Fig. 2, these additional scores are combined with the first 5 ones of each agent: these 20 scores are represented, as well as the barplot for each agent. The test statistics are computed using these 20 scores. Again, the difference is not big enough to make a decision given the available information, so we make a third interim: each agent is executed 5 more times. At the bottom of Figure 2, these 30 scores are represented, as well as the barplot for each agent. The test statistics are

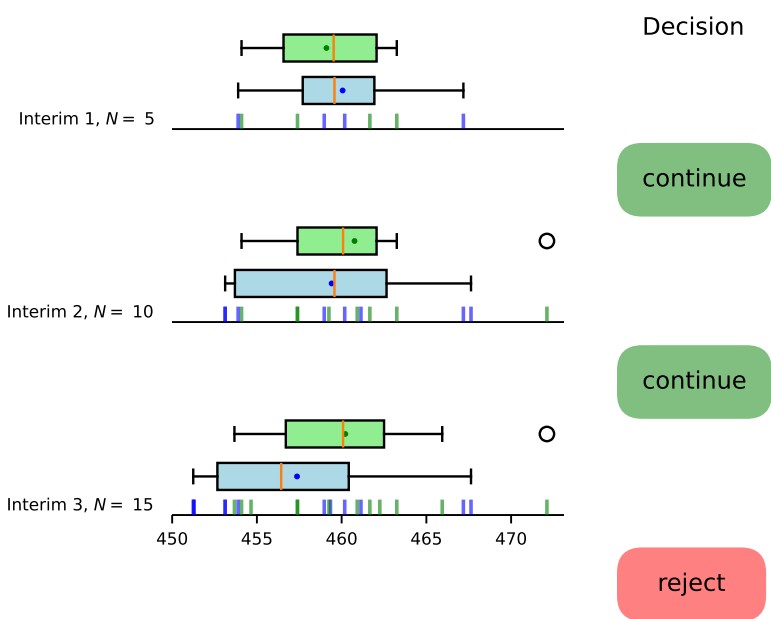

Figure 2: Illustration of the sequential and adaptive aspects of ADASTOP.

computed again on these 15 scores per agent. Now, ADASTOP decides to reject the null hypothesis, which means that the two set of scores of the two agents are indeed different, and terminates: the compared agents do not perform similarly. As explained in Section 2.5, this shows that the green agent performs most likely better than the blue one.

### 2.3 Current approaches for RL agents comparison

In the RL community, different approaches currently exist to compare agents. In (Colas et al., 2018; 2019), the authors show how to use hypothesis testing to test the equality between agents. Compared to our work, their approach is non-adaptive and only compares two agents. In (Patterson et al., 2023), the authors explore a similar workflow with added steps specialized to deep RL (hyperparameter optimization, choice of the testing environment...). Another line of works can be found in (Agarwal et al., 2021) in which the authors compare many agents using confidence intervals.

In this section, we summarize some of the problems we identify with the current approaches used to compare two or more RL agents in research articles.

**How many scores should we use?**  The number of scores used in practice in RL is quite arbitrary and often quite small (see Figure 1a). An intuition comes from the law of large numbers. As the performance of an agent is represented by the true mean of its scores, the more scores, the more precise the estimation of its performance. However, this does not tell us anything about what is a sufficient number of scores to draw a statistically significant conclusion.

**Theoretically sound comparison of multiple agents.**

According to statistical theory, in order to compare more than 2 agents, we need more samples from each agent than when we compare only two agents. The basic idea is that there is a higher chance to make an error when we perform multiple comparisons than when we compare only two agents, hence we need more data to have a lower probability of error at each comparison. This informal argument is formalized in the theory of multiple testing. However, the theory of multiple testing has almost never been used to compare

RL agents (with the notable exception of Patterson et al. (2023, Section 4.5)). In this paper, we remedy this with ADASTOP giving a theoretically sound workflow to compare 2 or more agents.

**Theoretically sound study when comparing agents on a set of tasks.**

Atari environments (Bellemare et al., 2013) are famous benchmarks in Deep RL. Due to time constraints, when using these environments, it is customary to use very few scores for one given game (typically 3 scores) and compare the agents on many different games. The comparisons are then aggregated: agent $A_1$ perform better than agent $A_2$ on more than 20 games out of the 26 games consider in the experiments. In terms of rigorous statistics, this kind of aggregation is complex to analyse properly because reward distributions are not the same in all games. $A_2$ may be better than $A_1$ only on some easy games: does this mean that $A_1$ is better than $A_2$? Up to our knowledge, there is not any proper statistical guarantee for this kind of comparison.

Advances have been made in (Agarwal et al., 2021) to interpret and visualize the results of RL agents in Atari environments. In particular the authors advise plotting confidence intervals and using the interquartile mean instead of the mean as aggregation functions. Correctly aggregating the comparisons on several games in Atari is still an *open problem*, and it is *beyond the scope of this article*. In this article, we suppose that we compare the agents on a single task, and we leave the comparison on a set of different tasks for future work. A discussion on these methods and the challenges of aggregating the results from several Atari environments can be found in the Appendix G.

## 2.4   Some methodologies for comparison of RL agents

Figure 1 of (Patterson et al., 2023) defines a workflow for the meaningful comparison of two RL agents given an environment and a performance measure. In addition to the usual considerations regarding which statistics to compare (Colas et al., 2018; Agarwal et al., 2021), (Patterson et al., 2023) also include hyperparameter choice in the workflow. For that, they recommend to use 3 scores per algorithm per set of hyperparameters to identify a good choice of hyperparameters for a given algorithm. When this is done, a fixed number of scores (set using expert knowledge on the environment) are computed for each of these fully-specified agents. These scores are then used for statistical comparison. ADASTOP fits at the end of this workflow.

In Fig 3, we showcase the use of ADASTOP to compare SAC (Haarnoja et al., 2018) to other Deep RL algorithms on HalfCheetah and Hopper Mujoco tasks. One can imagine a scenario in which SAC inventors follow (Patterson et al., 2023) methodology. After finding the best hyperparameters for TRPO, PPO and DDPG (Schulman et al., 2015; 2017; Lillicrap et al., 2015) agents are compared with ADASTOP using a minimal number of scores to get significant statistics. (Patterson et al., 2023) recommends 15 scores per agent per environment for a maze environment but this number of scores should vary in other environments, and it is not clear how many scores should be used for HalfCheetah and Hopper. This is why we need to use ADASTOP.

Using ADASTOP, we collect scores in an adaptive manner. This allows us to state that we collected enough scores to conclude that the SAC agent is MLB than other agents on HalfCheetah, and MLB than DDPG and TRPO agents on Hopper. As in all statistical tests, the conclusion holds up to a certain confidence level. A more in-depth study of the agents performance on Mujoco environments is given in Section 5.3.

## 2.5   Limitations of AdaStop and comparison to other statistical tests

Before presenting the theory behind ADASTOP, we want to make it clear that doing a statistically sound comparison with very few samples is a hard problem, and ADASTOP is one way to partially answer this problem. In particular, (i) we do not claim that ADASTOP is optimal, and (ii) more work is still necessary to prove theoretical guarantees for ADASTOP to support the empirical performances exhibited in the experiments (see in particular Section 5.2 and discussions on the power of the test).

The main theoretical limitation of ADASTOP comes from the fact that ADASTOP is distribution-free: it does not make any assumption on the distribution of the scores except a finite variance to get the asymptotic guarantees. As a consequence, the only non-asymptotic guarantees that we have are based on the comparison

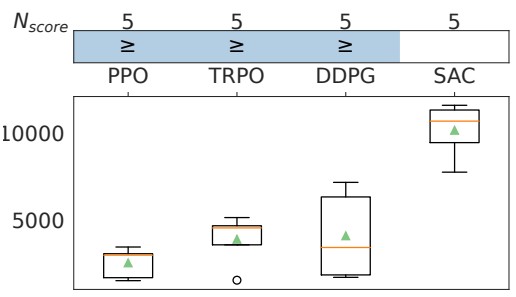
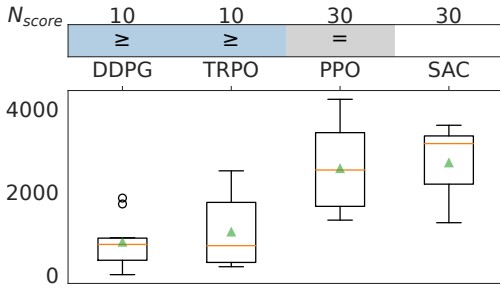

(a) Comparisons on HalfCheetah.

(b) Comparisons on Hopper.

Figure 3: Example of the use of ADASTOP to benchmark SAC in practice. We set the maximum number of runs $B$ of each agent to 30. The upper row tables represent the conclusions when comparing SAC to the agents in the column using $N_{score}$ scores. For example, on HalfCheetah, ADASTOP concludes that SAC is MLB than PPO using 5 scores for SAC and 5 scores for PPO. On Hopper, 10 scores are enough to conclude that SAC is MLB than DDPG and TRPO, and ADASTOP concludes that SAC and PPO perform equally using the maximum budget of $B$ scores for both SAC and PPO.

of score distributions and not on the comparison of their means. Formally, this means that we have control over the error (in Theorem 1, the error is shown to be equal to the parameter of the test $\alpha$) when doing the tests:

$$H_j : P_{l_1} = P_{l_2}, \quad \text{against} \quad H'_j : P_{l_1} \neq P_{l_2}$$

where $P_{l_i}$ is the distribution of scores for agent $l_i$, and $(l_1, l_2)$ are the indices of the agents we want to compare. This comes in contrast with traditional Gaussian tests like the t-test, which have strong theoretical guarantees under strong Gaussian assumptions (see Table 1 for a comparison of the guarantees of some classical tests, in particular for nonparametric tests, a finite variance is not sufficient to get strong non-asymptotic results).

Comparing distributions is not what we really want to do, but it is how to proceed when performing distribution-free tests and if the distributions concentrate sufficiently well (*i.e.*, have a finite variance) and the maximum sample size $N \times K$ is not too low. The Gaussian approximation of the sample mean justifies the way we use ADASTOP in this article. In particular, if we suppose a finite variance (a hypothesis that corresponds to a very weak concentration hypothesis necessary to get a central limit theorem), we give asymptotic guarantees on the comparison of the means (see Theorem 2):

$$H_j : \mathbb{E}_{P_{l_1}}[X] = \mathbb{E}_{P_{l_2}}[X], \quad \text{against} \quad H'_j : \mathbb{E}_{P_{l_1}}[X] \neq \mathbb{E}_{P_{l_2}}[X]. \tag{1}$$

Asymptotic guarantees remain unsatisfactory because we are targeting small sample sizes. Non-asymptotic guarantees are harder to obtain and would typically require concentration assumptions such as sub-Gaussian or bounded distributions; we leave such theoretical concerns for future works. In practice, this means that ADASTOP approximately compares the means for a large enough sample size, and the practitioner should be aware that **if the size of an interim $N$ is too small, AdaStop could give wrong results**. For this reason, we advise the practitioner to set $N$ and $K$ such that $N \times K \geq 30$. For a more precise study of the sample size effect, see Section 5.2. Please note that ADASTOP being an adaptive test, this does not mean that at least 30 scores have to be collected for each agent. This subtle point is illustrated in the experimental section of this paper.

Finally, note that the test expressed by Equation (1) is bidirectional, which means that in theory, concluding on this test does not tell us which of $P_{l_1}$ or $P_{l_2}$ has the largest mean score. Rather, it tells us that they are different. In practice, we use the sign of the difference in the empirical means to conclude which mean is larger. Using a bidirectional test and concluding on the direction afterward is often done by statisticians (see, in particular, the discussion in (Leventhal & Huynh, 1996) and the references therein), but this remains conceptually unsatisfactory. We keep the question of directional error for later work, which is why we do not

| Test | Hypothesis | Sequential | Theoretical Guarantees | | | |
| --- | --- | --- | --- | --- | --- | --- |
| | | | test $P = Q$ vs $P \neq Q$ | | test $\mu_P = \mu_Q$ vs $\mu_P \neq \mu_Q$ | |
| | | | Asymptotic | Non-Asymptotic | Asymptotic | Non-Asymptotic |
| t-test | Gaussian | ✗ | ✓ | ✓ | ✓ | ✓ |
| Wilcoxon | None | ✗ | ✓ | ✓ | ✗ | ✗ |
| Permutation | Finite variance | ✗ | ✓ | ✓ | ✓ | ✗ |
| Bootstrap | Finite variance | ✗ | ✓ | ✗ | ✓ | ✗ |
| Gaussian GST | Gaussian | ✓ | ✓ | ✓ | ✓ | ✓ |
| ADASTOP | Finite variance | ✓ | ✓ | ✓ | ✓ | ✗ |

Table 1: Comparison of the properties of several statistical methods. The Wilcoxon, Permutation and Bootstrap tests are nonparametric tests used in (Colas et al., 2019) for comparison in Deep RL and the Gaussian Group-Sequential test (Jennison & Turnbull, 1999) is a (parametric) sequential test often used in the context of clinical trials.

write that an agent performs "better" than another in this article but rather "most likely better (MLB)". See Section B.2 for a discussion on directional error.

## 3 Hypothesis testing to compare agent performance

In this section, we provide the background material on the statistical tests that we use to construct ADASTOP. First we review the statistical evaluation methods found in the literature, and then we describe our methodology, and we express some results on ADASTOP.

### 3.1 Literature overview of evaluation methods

In this section, we present some relevant references connected to statistical evaluation methodology.

**Nonparametric (and non-sequential) hypothesis testing.** One of our main challenges is to deal with the nonparametric nature of the data at hand. In the literature there has been a lot of works on nonparametric testing (see (Lehmann et al., 2005) for a comprehensive overview). Traditionally, the focus has been on asymptotic results due to challenges in deriving optimality for a nonparametric model. For most nonparametric tests, only rather weak (exact) theoretical results can be given, mostly on type I error (Romano, 1989; Shapiro & Hubert, 1979). Recent work on sequential nonparametric tests (Shin et al., 2021; Howard et al., 2021) show strong non-asymptotic results using concentration inequalities. However, these results often involve non-optimal constants, failing to explain non-asymptotic efficiency. In contrast, we use permutation tests due to their empirical (Ludbrook & Dudley, 1998) and theoretical (Kim et al., 2022) efficiency for small sample sizes.

**Sequential tests.** A closely related method for adaptive hypothesis testing consists in sequential tests. Two commonly used sequential tests are the Sequential Probability Ratio test (Wald, 1945) and the Generalized Likelihood Ratio test (Kaufmann & Koolen, 2021). In sequential testing, the scores are compared one after the other in a completely online manner. This is not adapted to our situation because in RL practice, one often trains several agents in parallel, obtaining a batch of scores at once. This motivates the use of group sequential tests (Jennison & Turnbull, 1999).

**Parametric group sequential tests**. In traditional hypothesis testing, data is analysed as a whole once it has been collected. Conversely, in a Group Sequential Test (GST) data are collected sequentially, the tests being performed at *interim* time points. At each interim, a new set of $N$ scores is collected (see (Jennison & Turnbull, 1999; Gordon Lan & DeMets, 1983; Pocock, 1977; Pampallona & Tsiatis, 1994) for references on GST). GST are often used in clinical trials to minimize the amount of data needed to conclude and this makes them well adapted for our purpose. The decision to continue sampling or conclude (with a controlled probability of error) depends on pre-defined stopping criteria used to define the tests. GST often makes strong assumptions on the data. In particular, it is often assumed that the data are i.i.d.and drawn from a Gaussian distribution (Jennison & Turnbull, 1999). This contrasts with our approach which is nonparametric.

**Bandits (Best arm identification or ranking).** Our objective is close to the one of bandit algorithms (Lattimore & Szepesvári, 2020): we *minimize the stopping time* (as in the fixed-confidence setting) of the test, and we have a *fixed maximum budget* (as in a fixed-budget setting). In our test, we allow a type I error with probability $\alpha \in (0, 1)$, which is similar to the fixed confidence setting while still having a fixed budget. In practice, our approach is more sample efficient than fixed budget bandit algorithms because these algorithms will always exhaust their budget and thereby achieve lower error rates. In contrast, ADASTOP allows larger error rate in exchange to higher sample efficiency.

### 3.2 Background material on the building-blocks of AdaStop

This section describes the basic building blocks used to construct ADASTOP: group sequential testing, permutation tests, and step-down method for multiple hypothesis testing. We explain these items separately, and then we combine them to create ADASTOP in Section 4. We also provide a small recap on hypotheses testing in the Appendix B for readers unfamiliar with these notions.

In order to perform the minimal number of runs, we propose to use group sequential testing (GST) with a nonparametric approach using permutation tests. Our approach is similar to (Mehta et al., 1994) but for multiple hypothesis testing. Compared to GST, we keep the i.i.d. assumption, but we do not assume that the data are drawn from a specific family of parametric distribution. In (Mehta et al., 1994), the authors use rank tests with group-sequential testing. In contrast with our work, (Mehta et al., 1994) does not provide theoretical guarantees and considers only the case of 2 agents.

#### 3.2.1 Permutation tests

Permutation tests are **nonparametric tests** that are exact for testing the equality of distributions. This means that the type I error of the test (*i.e.* the probability to make a mistake and reject the equality of two agents when their scores are statistically the same) is controlled by the parameter of the test $\alpha$, and that this is true for any fixed sample size $N$. Permutation tests are also well-known to work well in practice on **very small sample sizes** and are used extensively in biology. They were originally introduced by (Pitman, 1937) and (Fisher, 1936) (see (Lehmann et al., 2005, Chapter 17) for a textbook introduction). More recently (Chung & Romano, 2013) have studied asymptotic properties of this class of tests, while (Romano & Wolf, 2003) have focused on stepdown methods for multiple hypothesis testing.

Let us recall the basic formulation of a two-sample permutation test. Let $X_1, \dots, X_N$ be i.i.d. sampled from a law $P$ and $Y_1, \dots, Y_N$ i.i.d. sampled from a law $Q$. We want to test $P = Q$ against $P \neq Q$. Let $Z_i = X_i$ if $i \leq N$ and $Z_i = Y_{i-N}$ if $i > N$, $Z_1, \dots, Z_{2N}$. The test proceeds as follows: we reject $P = Q$ if $T(\text{id}) = \left| \frac{1}{N} \sum_{i=1}^{N} (Z_i - Z_{N+i}) \right|$ is larger than a proportion $(1 - \alpha)$ of the values $T(\sigma) = \left| \frac{1}{N} \sum_{i=1}^{N} (Z_{\sigma(i)} - Z_{\sigma(N+i)}) \right|$ where $\sigma$ enumerates all possible permutations of $\{1, \dots, 2N\}$ and id is the identity permutation ($\text{id}(i) = i$ for all $i$). Formally, we define the $(1 - \alpha)$-quantile as

$$B_N = \inf \left\{ b > 0 : \frac{1}{N!} \sum_{\sigma \in \mathbf{S}_N} \mathbb{1}\{T(\sigma) \geq b\} \leq \alpha \right\}$$

and we reject $P = Q$ when $T(\text{id}) \geq B_N$. The idea is that if $P \neq Q$, then $T(\text{id})$ should be large, and due to compensations, most $T(\sigma)$ should be smaller than $T(\text{id})$. Conversely, if $P = Q$, the difference of mean $T(\sigma)$ will be closer to zero. It is then sufficient to compute $T(\text{id})$ and $B_N$ in order to compute the decision of the test. Please note that this is a fairly usual simplification in the nonparametric tests literature to test the equality in distribution instead of the equality of the mean, because equality between distributions is easier to deal with[1]. It can be shown that permutation tests are nonetheless a good approximation of doing a comparison on the means (see Appendix E).

---

[1]In a statistical test, we want to have control on the error when $H_0$ is true. $H_0$ can be seen as asserting that the distribution that generated the data is in a certain set of distributions $\mathcal{P}_0$. The larger $\mathcal{P}_0$, the more complicated it is to make a statistical test without using strong assumption on the data. When comparing distributions, $\mathcal{P}_0 = \{P, Q \text{ probability distributions } | P = Q\}$, while when comparing the means, $\mathcal{P}_0$ is much larger because $\mathcal{P}_0 = \{P, Q \text{ probability distributions } | \mathbb{E}_P[X] = \mathbb{E}_Q[X]\}$.

### 3.2.2 GST comparison of two agents

In this section, we compare two agents $A_1$ and $A_2$ through a group sequential test that can be seen as a particular case of ADASTOP for two agents (see Section 4). In this simplified case, the testing procedure is presented in Algorithm 1. We leave the case with more than 2 agents to compare and the full version of ADASTOP, including multiple hypothesis testing, for Section 4. Algorithm 1 uses a permutation test where, at each interim, the boundary deciding the rejection is derived from the permutation distribution of the difference of empirical means observed across all previously obtained data. In what follows, $N$ to denote the number of scores collected at interim $k$.

We denote by $\mathbf{S}_{2N}$ the set of permutations of $\{1, \ldots, 2N\}$, $\sigma \in \mathbf{S}_{2N}$ one permutation and $\sigma(n)$ the $n$-th element of $\sigma$ for $n \in \{1, \ldots 2N\}$. In the GST setting, we perform a permutation test at each interim $k$, and $\sigma_k \in \mathbf{S}_{2N}$ denotes the permutation at interim $k$. For $\sigma_1, \sigma_2, \ldots, \sigma_k \in \mathbf{S}_{2N}$, we denote $\sigma_{1:k} = \sigma_1 \cdot \sigma_2 \cdot \ldots \cdot \sigma_k$ the concatenation[2] of the permutation $\sigma_1$ done in interim 1 with $\sigma_2$ done on interim 2,..., and $\sigma_k$ on interim $k$. Then, $e_{\sigma_i(n),i}$ denotes the score corresponding to the $n$-th element of the permuted sample at interim $i$, permuted by $\sigma_i$ (in the notations of Section 3.2.1 this corresponds to $Z_{\sigma(i)}$ but now, we specify the interim number in the notation). We denote:

$$T_{N,k}(\sigma_{1:k}) = \left| \sum_{i=1}^{k} \left( \sum_{n=1}^{N} e_{\sigma_i(n),i} - \sum_{n=N+1}^{2N} e_{\sigma_i(n),i} \right) \right|, \tag{2}$$

and the decision boundary:

$$B_{N,k} \in \inf \left\{ b > 0 : \frac{1}{((2N)!)^k} \sum_{\sigma_{1:k} \in \widehat{\mathcal{S}}_{N,k}} \mathbb{1}\{T_{N,k}(\sigma_{1:k}) \geq b\} \leq \frac{\alpha}{K} \right\}, \tag{3}$$

where $K$ is the total number of interims and $\widehat{\mathcal{S}}_{N,k}$ is the set of permutations $\sigma_{1:k} \in (\mathbf{S}_{2N})^k$ such that the test was not rejected before interim $k$, *e.g.*

$$\widehat{\mathcal{S}}_{N,k} = \{\sigma_{1:k} \in (\mathbf{S}_{2N})^k : \quad \forall m < k, \quad T_{N,m}(\sigma_{1:m}) \leq B_{N,m}\}. \tag{4}$$

One can show that this is equivalent to choosing $B_{N,k}$ such that

$$\mathbb{P}_{\sigma_{1:k}} \left( T_{N,m}(\sigma_{1:m}) > B_{N,k}, \forall m < k, \quad T_{N,m}(\sigma_{1:m}) \leq B_{N,m} \right) \leq \frac{\alpha}{K}.$$

We will see in Theorem 1 that this allows us to have control on the type I error of the test.

### 3.2.3 Multiple hypothesis testing

In order to compare more than two agents we need to perform multiple comparisons. This calls for a **multiple simultaneous statistical tests** (Lehmann et al., 2005, Chapter 9). The idea is that the probability to mistakenly reject a null hypothesis (type I error) generally applies only to each test considered individually. On the other hand, in order to conclude on all the tests at once, it is desirable to have an error controlled over the whole family of simultaneous tests. For this purpose, we use the family-wise error rate (Tukey, 1953) which is defined as the probability of making at least one type I error.

**Definition 1** (Family-Wise Error (Tukey, 1953)). *Given a set of hypothesis $H_j$ for $j \in \{1, \ldots, J\}$, its alternative $H_j'$, and $\boldsymbol{I} \subset \{1, \ldots, J\}$ the set of the true hypotheses among them, then the family-wise error (FWE) is defined by:*

$$\text{FWE} = \mathbb{P}_{H_j, j \in \boldsymbol{I}} (\exists j \in \boldsymbol{I} : \quad reject\ H_j),$$

*where $\mathbb{P}_{H_j, j \in \boldsymbol{I}}$ denotes the probability distribution for which all hypotheses $j \in \boldsymbol{I}$ hold true[3]. We say that an algorithm has a weak FWE control at a joint level $\alpha \in (0, 1)$ if the FWE is smaller than $\alpha$ when all the hypotheses are true, that is $\boldsymbol{I} = \{1, \ldots, J\}$ but not necessarily otherwise. We say it has strong FWE control if FWE is smaller than $\alpha$ for any non-empty set of true hypotheses $\boldsymbol{I} \neq \emptyset$.*

---

[2]Here, the word "concatenation" means that we apply $\sigma_1$ on $\{1, \ldots, N\}$, then $\sigma_2$ on $\{N + 1, \ldots, 2N\}$ and so on, so that $\sigma_{1:k}(Nj + k) = \sigma_j(k)$.

[3]See also Appendix B for further explanations on this concept.

---

**Algorithm 1:** Adaptive stopping to compare two RL agents. This algorithm is expressed in the context of the comparison of RL agents. It is easy to adapt to other types of computational agents.

---

**Parameters:** Agents $A_1, A_2$, environment $\mathcal{E}$, number of interims $K \in \mathbb{N}^*$, size of an interim $N$, error parameter $\alpha \in (0,1)$.

**1 for** $k = 1, \ldots, K$ **do**
**2**   **for** $l = 1,2$ **do**
**3**    Train agent $A_l$ on environment $\mathcal{E}$ N times, with the seeds $s_{l,(k-1)N+1}, \ldots, s_{l,kN}$. This generates
    $N$ policies $\pi_{l,(k-1)N+1}, \ldots, \pi_{l,kN}$
**4**    Collect scores $e_{1,k}(A_l), \ldots, e_{N,k}(A_l)$ by running each policy $\pi_{l,(k-1)N+1}, \ldots, \pi_{l,kN}$.
**5**   **end**
**6**   Compute the boundary $B_{N,k}$ using Equation (3).
**7**   **if** $T_{N,k}(\mathrm{id}) \geq B_{N,k}$ **then**
**8**    return reject
**9 end**
**10 return accept**

---

If we want to test the equality of $L$ distributions $P_1, P_2, \ldots, P_L$, the straightforward way is to do a pairwise comparison. This creates $J = \frac{L(L-1)}{2}$ hypotheses. We let $\mathbf{C} = \{\mathbf{c}_1, \ldots, \mathbf{c}_J\}$ be the set of all possible comparisons between the distributions, where $\mathbf{c}_j = (l_1, l_2) \in \mathbf{C}$ denotes a comparison between distributions $P_{l_1}$ and $P_{l_2}$ for $l_1, l_2 \in \{1, 2, \ldots, L\}$. Therefore, for $\mathbf{c}_j = (l_1, l_2)$, $H_{\mathbf{c}_j}$ denotes a hypothesis stating that $P_{l_1}$ and $P_{l_2}$ are equal and its alternative $H'_{\mathbf{c}_j}$ is that $P_{l_1}$ and $P_{l_2}$ are different. On the other hand, if one wants to compare an agent to agents whose ranking does not interest us, it may be sufficient to only compare this agent to all the others yielding the $L - 1$ comparisons $c_1 = (1, l_2)$ for $2 \leq l_2 \leq L$.

There are several procedures that can be used to control the FWE. The most famous one is Bonferroni's procedure (Bonferroni, 1936) recalled in the Appendix (Section B). As Bonferroni's procedure can be very conservative in general, we prefer a *step-down* method (Romano & Wolf, 2003) that performs better in practice because it implicitly estimates the dependence structure of the test statistics. The step-down method is detailed in the next section.

### 3.2.4 Step-down procedure

(Romano & Wolf, 2003) proposed the step-down procedure to solve a multiple hypothesis testing problem. It is defined as follows (for a *non group-sequential test*): for a permutation $\sigma \in \mathbf{S}_{2N}$ and for $(e_n(j))_{1 \leq n \leq 2N}$ the random variables being compared in hypothesis $j$, the permuted test statistic of hypothesis $j$ is defined by:

$$T_N^{(j)}(\sigma) = \left| \sum_{n=1}^{N} e_{\sigma(n)}(j) - \sum_{n=N+1}^{2N} e_{\sigma(n)}(j) \right|. \tag{5}$$

This test statistic is extended to any subset of hypothesis $\mathbf{C} \subset \{1, \ldots, J\}$ with the following formula:

$$\overline{T}_N^{(\mathbf{C})}(\sigma) = \max_{j \in \mathbf{C}} T_N^{(j)}(\sigma). \tag{6}$$

To specify the test, one compares $\overline{T}_N^{(\mathbf{C})}(\mathrm{id})$ to some threshold value $B_N^{(\mathbf{C})}$, that is: we accept all hypotheses in $\mathbf{C}$ such that $\overline{T}_N^{(\mathbf{C})}(\mathrm{id}) \leq B_N^{(\mathbf{C})}$. The threshold of the test $B_N^{(\mathbf{C})}$ is defined as the quantile of order $1 - \alpha$ of the permutation law of $\overline{T}_N^{(\mathbf{C})}(\sigma)$:

$$B_N^{(\mathbf{C})} = \inf \left\{ b > 0 : \left( \frac{1}{(2N)!} \sum_{\sigma \in \mathbf{S}_{2N}} \mathbb{1}\{\overline{T}_N^{(\mathbf{C})}(\sigma) \geq b\} \right) \leq \alpha \right\}. \tag{7}$$

In other words, $B_N^{(\mathbf{C})}$ is the real number such that an $\alpha$ proportion of the values of $\overline{T}_N^{(\mathbf{C})}(\sigma)$ exceeds it, when $\sigma$ enumerates all the permutations of $\{1, \ldots, 2N\}$. The permutation test is summarized in Algorithm 2.

---

**Algorithm 2:** Multiple testing by step-down permutation test.

---

**Parameters:** $\alpha \in (0,1)$
**Input:** $e_n(j)$ for $1 \leq n \leq 2N$ and $j \in \mathbf{C}_0 = \{\mathbf{c}_1, \ldots, \mathbf{c}_J\}$.

**1** Initialize $\mathbf{C} \leftarrow \mathbf{C}_0$.
**2 while** $C \neq \emptyset$ **do**
**3** $\quad$ Compute $T_N^{(\mathbf{C})}(\sigma)$ for every $j$ and every $\sigma$ using Equation (6).
**4** $\quad$ Compute $B_n^{(\mathbf{C})}$ using Equation (7).
**5** $\quad$ **if** $\overline{T}_N^{(C)}(\mathrm{id}) \leq B_N^{(C)}$ **then**
**6** $\quad\quad$ Accept all the hypotheses $H_j, j \in \mathbf{C}$ and exit the loop.
**7** $\quad$ **else**
**8** $\quad\quad$ Reject $H_{\mathbf{c}_{j_{\max}}}$ where $\mathbf{c}_{j_{\max}} = \arg\max_{\mathbf{c}_j \in \mathbf{C}} T_N^{(j)}(\mathrm{id})$.
**9** $\quad\quad$ Define $\mathbf{C} = \mathbf{C} \setminus \{\mathbf{c}_{j_{\max}}\}$
**10** $\quad$ **end**
**11 end**

---

Algorithm 2 is initialized with $\mathbf{C} = \mathbf{C}_0$ containing all the comparisons we want to test. Then, it enters a loop where the test decides to reject or not the most extreme hypothesis in $\mathbf{C}$, *i.e.* $H_{j_{\max}}$ where $j_{\max} = \arg\max_{j \in \mathbf{C}} T_N^{(j)}(\mathrm{id})$. Here, $\mathbf{C}$ is the current set of neither yet rejected nor accepted hypotheses. If the test statistic $T_N^{(j)}(\mathrm{id})$ for the most extreme hypothesis in $\mathbf{C}$ (*i.e.* $\overline{T}_N^{\mathbf{C}}(\mathrm{id})$) does not exceed the given threshold $B_n^{(\mathbf{C})}$, then all hypotheses in $\mathbf{C}$ are accepted, and the loop is exited. Otherwise, the most extreme hypothesis is discarded from the set $\mathbf{C}$ and another iteration is performed until either all remaining hypothesis are accepted, or the set of remaining hypotheses is empty.

The maximum of the statistic in Equation (6) for $\sigma = \mathrm{id}$ allows us to test intersections of hypotheses, while the threshold $B_n^{(\mathbf{C})}$, under the null hypotheses of equality of distribution, allows for strong control on the FWE (*i.e.* FWE $\leq \alpha$). This last result follows from (Romano & Wolf, 2003, Corollary 3) and is a particular case of Theorem 1 for ADASTOP. In fact, this procedure is not specific to permutation tests, and it can be used for other tests provided some properties hold on the thresholds $B_n^{(\mathbf{C})}$.

## 4 AdaStop: adaptive stopping for nonparametric group-sequential multiple tests

In this section, we present the construction and the theoretical properties of ADASTOP (see Algorithm 3). ADASTOP compares the scores of multiple agents in an *adaptive* rather than a fixed way. We consider $L \geq 2$ agents $A_1, \ldots, A_L$. As above, we let $\mathbf{C}_0 = \{\mathbf{c}_1, \ldots, \mathbf{c}_J\} \subseteq \{1, \ldots, L\}^2$ be the set of all the comparisons to make between the agents. $\mathbf{I}$ denotes the set of indices of the true hypotheses among $\{1, \ldots, J\}$.

Algorithm 3 specifies ADASTOP. ADASTOP relies on the test statistic $\overline{T}_{N,k}^{(\mathbf{C})}(\sigma_{1:k})$ defined in equation (9), and the boundary thresholds $B_{N,k}^{\mathbf{C}}$ defined in Equation (10). We discuss a few implementation details in the rest of this section.

*Definition of the test statistic.* Let $e_{1,i}(j), \ldots, e_{2N,i}(j)$ denote the $2N$ scores used at interim $i$. They are obtained through the execution of the policies resulting from the training of the two agents $A_{l_1}$ and $A_{l_2}$ for which $\mathbf{c}_j = (l_1, l_2)$. We also consider permutations of these scores to define the test statistics $T_{N,k}^{(j)}$ below. For a comparison $j$, we consider a permutation $\sigma_i \in \mathbf{S}_{2N}$ at interim $i$ that reshuffles the order of the scores mapping $n \in \{1, \ldots, 2N\}$ on $\sigma_i(n) \in \{1, \ldots, 2N\}$. Note that if $n \in \{1, \ldots, N\}$ and $\sigma_i(n) \in \{N+1, \ldots, 2N\}$, we are exchanging a score of the first agent with a score of the second agent in the comparison. It can also happen that instead, we permute two scores of the same agent. The difference between the two cases is

important for the definition of the following permutation statistic:

$$T_{N,k}^{(j)}(\sigma_{1:k}) = \left| \sum_{i=1}^{k} \left( \sum_{n=1}^{N} e_{\sigma_i(n),i}(j) - \sum_{n=N+1}^{2N} e_{\sigma_i(n),i}(j) \right) \right|. \tag{8}$$

In other words, $T_{N,k}^{(j)}(\sigma_{1:k})$ is the absolute value of the sum of differences of all scores until interim $k$ after consecutive permutations of the concatenation of the two agents scores by $\sigma_1, \dots, \sigma_k \in \mathbf{S}_{2N}$. Let $\mathbf{C} \subseteq \mathbf{C}_0$ be a subset of the set of considered hypothesis and let us denote:

$$\overline{T}_{N,k}^{(\mathbf{C})}(\sigma_{1:k}) = \max_{j \in \mathbf{C}} T_{N,k}^{(j)}(\sigma_{1:k}), \tag{9}$$

$\overline{T}_{N,k}^{(\mathbf{C})}(\sigma_{1:k})$ is the test statistic used in ADASTOP. The construction of the test is inspired by the permutation tests of Equation (5) used to test intersection of hypotheses as in Equation (6) in the step-down method presented in Section 3.2. Still, it also incorporates group sequential tests from Section 3.2.2 and its test statistic introduced in Equation (2).

*Choice of permutations.* Instead of using all the permutations as we did for now, one may use a random subset among all permutations $\mathcal{S}_{N,k} \subset \{\sigma_{1:k}, \quad \forall i \leq k, \sigma_i \in \mathbf{S}_{2N}\}$ to speed-up computations. The theoretical guarantees persist as long as the choice of the permutations is made independent on the data. Using a small number of permutations will decrease the total power of the test, but with a sufficiently large number of random permutations (typically for the values of $N$ and $K$ we consider, $10^4$ permutations are sufficient) the loss in power is acceptable. We need to include the identity in addition to the random permutations to keep the type I error guarantee (Phipson & Smyth, 2010).

$T_{N,k}$ does not change when the permutation does not exchange any index from $\{1, \dots, N\}$ with an index of $\{N+1, \dots, 2N\}$. In essence, choosing a permutation is equivalent to choosing the signs in $\sum_{n=1}^{N} e_{\sigma_i(n),i}(j) - \sum_{n=N+1}^{2N} e_{\sigma_i(n),i}(j)$. And because we take the absolute value, there are $\frac{1}{2}\binom{2N}{N}$ possible unique values to $T_{N,1}$ (up to ties when the score distributions are discrete). Then, by enumerating all the permutations for the other interims, there are $\frac{1}{2}\binom{2N}{N}^k$ possible unique values to $T_{N,k}$.

In practice, we use a parameter $B \in \mathbb{N}$ and the number of permutations used at interim $k$ will be $|\mathcal{S}_{N,k}| = m_k = \min\left(B, \frac{1}{2}\binom{2N}{N}^k\right)$, *i.e.* whenever possible, we use all the permutations and if this is too much, we use permutations drawn at random.

*Definition of the boundaries.* With these permutations, we define the boundary thresholds $B_{N,k}^{(\mathbf{C})}$ by:

$$B_{N,k}^{(\mathbf{C})} = \inf \left\{ b > 0 : \frac{1}{m_k} \sum_{\sigma \in \widehat{\mathcal{S}}_{N,k}} \mathbb{1}\{\overline{T}_{N,k}^{(\mathbf{C})}(\sigma_{1:k}) \geq b\} \leq q_k \right\}. \tag{10}$$

where $\sum_{j=1}^{k} q_j \leq \frac{k\alpha}{K}$ and where $\widehat{\mathcal{S}}_{N,k}$ is the subset of $\mathcal{S}_{N,k}$ such that the statistic associated to the permutation would not have been rejected before. Formally, $\widehat{\mathcal{S}}_{N,k}$ is the following set of permutations:

$$\widehat{\mathcal{S}}_{N,k} = \left\{ \sigma_{1:k} : \forall m < k, \quad \overline{T}_{N,m}^{(\mathbf{C})}(\sigma_{1:m}) \leq B_{N,m}^{(\mathbf{C})} \right\}. \tag{11}$$

Note that $q_1$ is not equal to $\alpha/K$. Due to discreteness (we use an empirical quantile over a finite number of values), $q_1$ is chosen equal to $\lfloor \frac{\alpha}{2K}\binom{2N}{N} \rfloor / (\frac{1}{2}\binom{2N}{N})$. Similarly, $q_2$ is chosen to be as large as possible while having $q_1 + q_2 \leq 2\alpha/K$. And so on and so forth: in practice this means taking the $q_i$'s as follows:

$$q_i = \lfloor \tfrac{\alpha i}{2K}\binom{2N}{N} \rfloor / (\tfrac{1}{2}\binom{2N}{N}) - q_{i-1} \text{ for } 2 \leq i \leq K, \text{ and } q_0 = 0. \tag{12}$$

One may notice that due to the discrete nature of the permutation distribution, the confidence of the test performed by ADASTOP is not $\alpha$ but the sum of the $q_i$'s which may be smaller than the prescribed $\alpha$.

---

**Algorithm 3:** ADASTOP (main algorithm) in the context of the comparison of 2 RL agents. Its application to other types of computational agents is straightforward.

---

**Parameters:** Agents $A_1, A_2, \ldots, A_L$, environment $\mathcal{E}$, comparison pairs $(c_j)_{j \leq J}$ where $c_i$ is a couple of agents that we want to compare. Integers $K, N \in \mathbb{N}^*$, test parameter $\alpha$.

**1** Set $\mathbf{C} = \{1, \ldots, J\}$ the set of indices for the comparisons to perform.

**2 for** $k = 1, \ldots, K$ **do**

**3**      **for** $l = 1 \ldots L$ **do**

**4**          Train $N$ times agent $A_l$ on environment $\mathcal{E}$. This generates $N$ policies.

**5**          Collect the $N$ scores of the $N$ policies.

**6**      **end**

**7**      **while** *True* **do**

**8**          Compute the boundaries $B_{N,k}^{(\mathbf{C})}$ using Equation (10).

**9**          **if** $T_{N,k}^{(C)}(\mathrm{id}) > B_{N,k}^{(C)}$ **then**

**10**              Reject $H_{j_{\max}}$ where $j_{\max} = \arg\max\left(\overline{T}_{N,k}^{(j)}(\mathrm{id}), \quad j \in \mathbf{C}\right)$.

**11**              $\mathbf{C} = \mathbf{C} \setminus \{j_{\max}\}$

**12**          **else**

**13**              Exit the while loop.

**14**          **end**

**15**      **end**

**16**      **if** $C = \emptyset$ **then** Exit the loop and return the decision of the test.

**17**      **if** $k = K$ **then** Exit the loop and accept all hypotheses remaining in $\mathbf{C}$.

**18 end**

---

### 4.1 Theoretical guarantees of AdaStop

One of the basic properties of two-sample permutation tests is that when the null hypothesis is true, then all permutations are as likely to give a certain value and permuting the sample should not change the test statistic too much. Following our choice of $\overline{B}_{N,k}$ as a quantile of the law given the data, the algorithm has a probability to wrongly reject the hypothesis bounded by $\alpha$. This informal statement is made precise in the following theorem.

**Theorem 1** (Controlled family-wise error)**.** *Let $\alpha \in (0,1)$, and consider the multiple testing problem $H_j : P_{l_1} = P_{l_2}$ against $H'_j : P_{l_1} \neq P_{l_2}$ for all the couples $\mathbf{c}_j = (l_1, l_2) \in \{\mathbf{c}_1, \ldots, \mathbf{c}_J\}$. Then, the test resulting from Algorithm 3 has a strong control on the family-wise error for the multiple test, i.e. if we suppose that all the hypotheses $H_i, i \in \mathbf{I}$ are true and the others are false, then*

$$\mathbb{P}\left(\exists j \in \mathbf{I}: \quad \text{reject } H_j\right) \leq \alpha.$$

The proof of Theorem 1 is given in the Appendix (Section C).

*Hypotheses of the test*: in Theorem 1 we show that Algorithm 3 tests the equality of the distributions $P_{l_1} = P_{l_2}$ versus $P_{l_1} \neq P_{l_2}$, whereas in practice we would prefer to compare the means of the distribution $\mu_{l_1} = \mu_{l_2}$ versus $\mu_{l_1} \neq \mu_{l_2}$, see Section 2.5 for a discussion on the differences between the two. We show in the Appendix E that for large $N$, the test comparing the means $\mu_{l_1} = \mu_{l_2}$ versus $\mu_{l_1} \neq \mu_{l_2}$ has the right guarantees (FWE smaller than $\alpha$). This shows that even though we test the distributions, we also have an approximate test on the means. More precisely, we show the following in the Appendix E for the comparison of the means of two distributions.

**Theorem 2.** *Suppose that $\alpha \in (0,1)$, suppose that $P$ and $Q$ both have a finite variance, and consider the two-sample testing problem $H_0 : \mathbb{E}_P[X] = \mathbb{E}_Q[X]$ against $H'_0 : \mathbb{E}_P[X] \neq \mathbb{E}_Q[X]$. Then, the test resulting from Algorithm 3 has an asymptotic type I error of $\alpha$*

$$\lim_{N \to \infty} \mathbb{P}_{H_0}\left(\text{reject } H_0\right) = \alpha.$$

The proof of Theorem 2 is given in the Appendix (Section E).

*Power of the test*: in Theorem 1, there is no information on the power of the test. For any $N, K$, we show that, the FWE is upper-bounded by $\alpha$. In the appendix E, we establish Theorem 3 that can be used to get a control on the asymptotic power. On the other hand, having non-asymptotic information on the power would allow us to give a rule for the choice of $N$ and $K$. However, non-asymptotic power analysis in a nonparametric setting is in general hard, and it is beyond the scope of this article. Instead, we compute empirically the power of our test and show that it performs well empirically compared to non-adaptive approaches. See Section 5.2 for the empirical power study on a Mujoco environment.

# 5    Experimental study

This section demonstrates ADASTOP from a practitioner perspective. First, we illustrate the statistical properties of ADASTOP on toy examples in which the scores of the agents are sampled from known distributions. Then, we compare empirically ADASTOP to non-adaptive approaches. Finally, we exemplify the use of ADASTOP on a real case to compare a set of Deep RL agents.

To reproduce the experiments of this paper, the python code is freely available on GitHub at `https://github.com/TimotheeMathieu/Adaptive_stopping_MC_RL`.

## 5.1    Toy examples

To start with, let us consider a toy example. In what follows, let us denote: (i) $\mathcal{N}(\mu, \sigma^2)$ the normal distribution with mean $\mu$ and standard deviation $\sigma$, (ii) $\mathcal{M}_f^{\mathcal{N}}(\mu_1, \sigma_1^2; \mu_2, \sigma_2^2)$ the mixture of 2 Gaussian distributions $\mathcal{N}(\mu_1, \sigma_1^2)$ and $\mathcal{N}(\mu_2, \sigma_2^2)$ with weights $f$ and $1 - f$ respectively.

We compare two agents $A_1$ and $A_2$ for which we know the distributions of their scores. We consider three cases in Fig. 4. In the 3 cases, the scores of agent $A_1$ are drawn from $\mathcal{N}(0, 0.01)$ whereas the scores of the agent $A_2$ are drawn from a certain mixture detailed below. We use a hyperparameter $\Delta$ which is the distance between the two modes of the mixture ($\Delta = |\mu_1 - \mu_2|$). Our goal is to compare the performance of agent $A_1$ with the performance of agent $A_2$ and we study the outcome of ADASTOP when $\Delta$ is increasing from 0 to 1. In the first case (upper-left part), the scores of $A_2$ are drawn from $\mathcal{M}_{\frac{1}{2}}^{\mathcal{N}}(-\Delta/2, 0.01; \Delta/2, 0.01)$: in this situation, the means of both distributions are both equal to 0, but the distributions are different (except when $\Delta = 0$). In the second case (upper-right part), the scores of $A_2$ are drawn from $\mathcal{M}_{\frac{1}{2}}^{\mathcal{N}}(0, 0.01; \Delta, 0.01)$: in this situation, as $\Delta$ increases, the means of the two distributions get more and more apart making the rejection of the null hypothesis easier and easier. In the third case (lower part), the scores of $A_2$ are drawn from $\mathcal{M}_{0.9}^{\mathcal{N}}(-0.1\Delta, 0.01; 0.9\Delta, 0.01)$: in this situation, the distributions are different but their means are both 0. In all three cases, we run ADASTOP with $K = 5$, $N = 5$ and $\alpha = 0.05$. We also limit the maximum number of permutations to $B = 10^4$. At the bottom of each subfigure of Figure 4, a color bar indicates the rejection rate of the null hypothesis that the compared distributions are the same for $\Delta \in [0, 1]$. By varying $\Delta$ from 0 to 1, we observe the evolution of the power of the test, *i.e.* the probability of rejecting the null hypothesis when it is indeed false. Figure 4a shows that the error of the test remains around 0.05 for all $\Delta$ (it is at most 0.1 for the most extreme case). Indeed, even though the distributions are different, their means remain the same. If the null hypothesis states that the means are the same, then ADASTOP will return the correct answer with type I error not larger than 0.095 (see Figure 4a) for $\alpha = 0.05$, which is larger than $\alpha$ due to the fact that the test for comparison of the means is asymptotic. This is an illustration of the fact that in addition to performing a test on the distributions, ADASTOP approximates the test on the means as shown theoretically in the asymptotic result in Appendix E and as discussed at the end of Section 4.1. In contrast, Figure 4b demonstrates an increasing trend, reaching a confidence level close to 1 when $\Delta > 0.6$, which corresponds to the case where the two modes are separated by 3 standard deviations from both sides. Finally, in Figure 4c, we use a distribution made of an unbalanced mixture of 2 normal distributions. This is meant to model an agent that does not perform optimally except in some rare cases (*e.g.* the agent has a score around 0 for 90% of the evaluation runs). As in case 1, the mean of the scores for both agents is 0 and ADASTOP accepts the equality most of the time, with an error of 0.2 in the most extreme case which we think is acceptable given the difficulty of this setting.

To obtain an estimation of the error, we have executed each comparison $M = 5 \cdot 10^3$ times, and we plot confidence intervals corresponding to $3\sigma/\sqrt{M}$ (confidence larger than 99%) where $\sigma$ is the standard deviation of the test decision. In addition to cases 1, 2 and 3, we also provide a fourth experiment with a comparison of 10 agents in Appendix H.1.

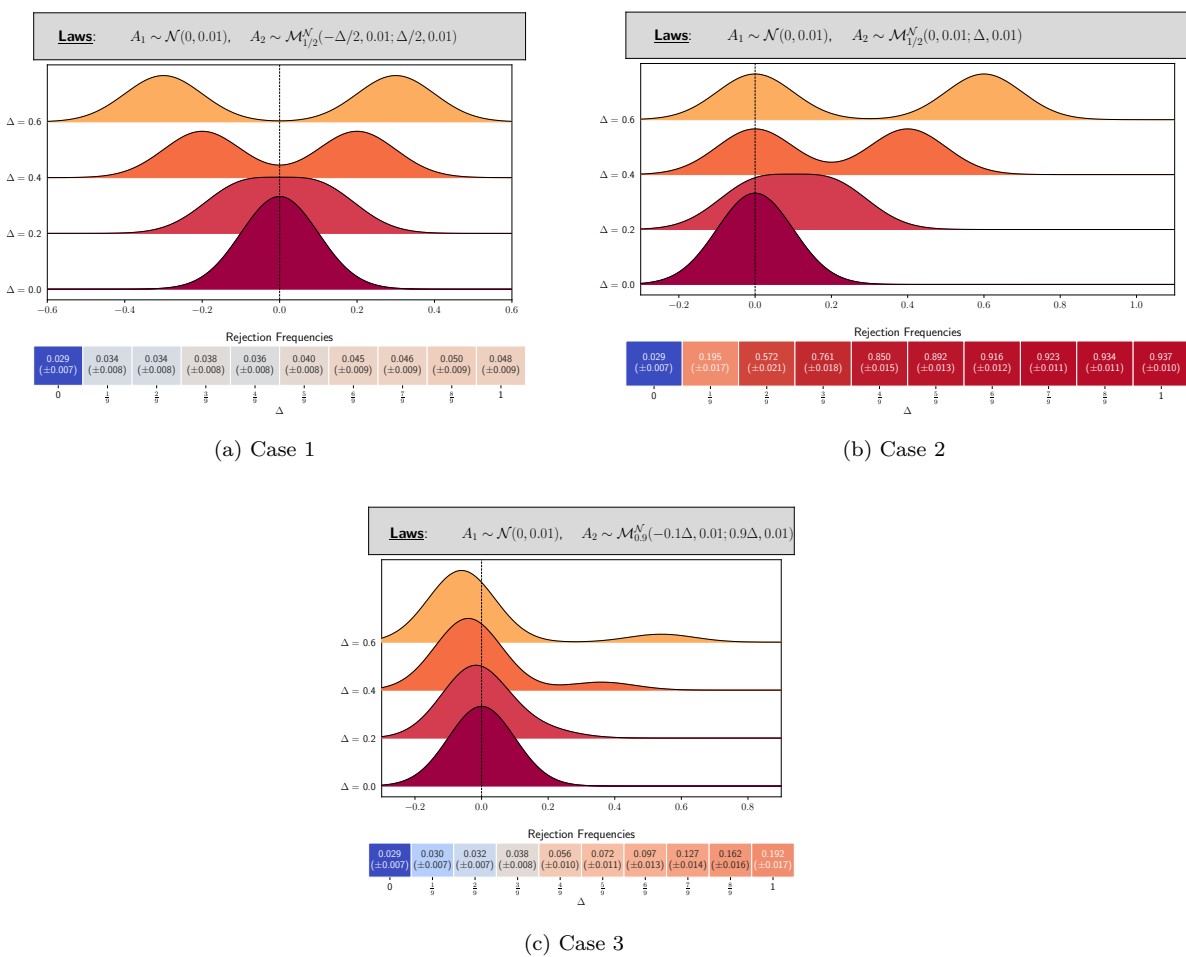

(a) Case 1

(b) Case 2

(c) Case 3

Figure 4: Toy examples. For each of these 3 cases, we plot the distribution of scores of each of the 2 agents that are compared, as well as the frequency of rejection of the null hypothesis. Please, refer to the text for details.

## 5.2 Comparison with non-adaptive approach

(Colas et al., 2019) share the same objective with us. However, they use non-adaptive tests unlike ADASTOP. We follow their experimental protocol and compare ADASTOP and non-adaptive approaches empirically in terms of statistical power as a function of the sample size (number of scores). In particular, we use the data[4] they provide for a SAC agent and for a TD3 agent evaluated on HalfCheetah (see Fig. 12 in the Appendix). Similarly to (Colas et al., 2019, Table 15), we compute the empirical statistical power of ADASTOP as a function of the number of scores of the RL agents (Table 2). To compute the empirical statistical power for a given number of scores, we make the hypothesis that the distribution of SAC and TD3 agents scores are different, and we count how many times ADASTOP decides that one agent is MLB than the other (number of true positives). As the test is adaptive, we also report the effective number of scores that are necessary

---

[4]available at `https://github.com/flowersteam/rl_stats/tree/master/data`.

| N\K | 2 | 3 | 4 | 5 | 6 |
|-----|-----|-----|-----|-----|-----|
| 1 | 0.0 (2.0) | 0.0 (3.0) | 0.277 (4.0) | 0.465 (5.0) | 0.56 (6.0) |
| 2 | 0.005 (4.0) | 0.33 (6.0) | 0.531 (6.96) | 0.602 (8.345) | 0.704 (9.198) |
| 3 | 0.213 (5.984) | 0.506 (8.085) | 0.627 (10.212) | 0.689 (11.02) | 0.785 (11.52) |
| 4 | 0.371 (7.616) | 0.611 (9.648) | 0.744 (11.7) | 0.82 (12.08) | 0.845 (13.89) |
| 5 | 0.465 (9.044) | 0.691 (11.031) | 0.78 (13.28) | 0.853 (14.27) | 0.884 (14.532) |
| 6 | 0.534 (10.4) | 0.73 (12.306) | 0.837 (14.124) | 0.89 (14.94) | 0.911 (15.978) |
| 7 | 0.599 (11.358) | 0.779 (13.404) | 0.879 (14.916) | 0.92 (15.495) | 0.939 (16.404) |
| 8 | 0.635 (12.322) | 0.818 (13.95) | 0.885 (15.824) | 0.942 (16.03) | 0.961 (17.268) |

Table 2: Average empirical statistical power and, in parentheses, effective number of scores used by ADASTOP as a function of the total number of scores ($N \times K$) when comparing SAC and TD3 agents on Mujoco HalfCheetah task. The number of permutations $B$ is set to $10^4$ and $\alpha$ is set to 0.05. ADASTOP is run $10^3$ times for each ($N, K$) pair. The shades of blue are proportional to the power, a value in $[0, 1]$ (we use the same color scheme as in (Colas et al., 2018)). It can seem strange that the power is 0 in some cases. This situation is explained in the text.

to make a decision with 0.95 confidence level. For each number of scores, we have run ADASTOP $10^3$ times. For example, when comparing the scores of SAC and TD3 on HalfCheetah using ADASTOP with $N = 4$ and $K = 5$, the maximum number of scores that is used is $N \times K = 20$. However, we observe in Table 2 that when $N = 4$ and $K = 5$, ADASTOP can make a decision with a power of 0.82 using only 12 scores. In (Colas et al., 2019, Table 15), the minimum number of scores required to obtain a statistical power of 0.8 when comparing SAC and TD3 agents is 15 when using either a t-test, or a Welch test, or a bootstrapping test. With this example, we first show that being an adaptive test, ADASTOP may save computations. We also show that as long as the scores of agents are made available, ADASTOP can use them to provide a statistically sound conclusion, and as such, ADASTOP may be used to assess the initial conclusions, hopefully strengthening them with a statistically significant argument.

*Remark*: as already mentioned above, when $N$ and $K$ are both small, it can happen that ADASTOP does not allow an error up to $\alpha$ but it will restrict itself to a smaller error and this will lead to a much lower power. For instance, if $N = 2, K = 2, \alpha = 0.05$ for the first interim, we have $\lfloor \frac{1}{2}\binom{2N}{N}\alpha/2 \rfloor = 0$ hence the first boundary is necessarily infinity and ADASTOP never rejects, *e.g.* with the notations of Equation (12), $q_1 = 0$. Then at the second interim $\lfloor \frac{1}{2}(\binom{2N}{N})^2\alpha/2 \rfloor = 3$ and we use $q_1 + q_2 = \lfloor \frac{1}{2}(\binom{2N}{N})^2\alpha/2 \rfloor/(\frac{1}{2}(\binom{2N}{N})^2) \simeq 0.041$ which is smaller than the 5% that we allow for the test, hence the test is more conservative than needed. This is an extreme case happening in the first few interims when $N$ is small explaining some values in Table 2 that may seem strange at first glance (for instance the power is equal to 0 when $N = 1$ and $K \le 3$).

### 5.3 AdaStop for Deep Reinforcement Learning

In this section, we use ADASTOP to compare four commonly-used Deep RL agents on the MuJoCo[5] (Todorov et al., 2012) benchmark for high-dimensional continuous control. We use the Gymnasium[6] implementation. More specifically, we train agents on the Ant-v3, HalfCheetah-v3, Hopper-v3, Humanoid-v3, and Walker-v3 environments using PPO from rlberry (Domingues et al., 2021), SAC from Stable-Baselines3 (Raffin et al., 2021), DDPG from CleanRL (Huang et al., 2022), and TRPO from MushroomRL (D'Eramo et al., 2021). PPO, SAC, DDPG, and TRPO are Deep RL algorithms used for continuous control tasks. We choose these algorithms because they are commonly used and they represent a diverse set of approaches from different RL libraries. We use different RL libraries in order to demonstrate the flexibility of ADASTOP, as well as to provide examples on how to use these popular libraries with ADASTOP.

On-policy algorithms, such as PPO and TRPO, update their policies based on the current data they collect during training, while off-policy algorithms, such as SAC and DDPG, can learn from any data, regardless of how it was collected. This difference may make off-policy algorithms more sample-efficient but less stable

---

[5]We use MuJoCo version 2.1, as required by https://github.com/openai/mujoco-py.
[6]https://github.com/Farama-Foundation/Gymnasium.

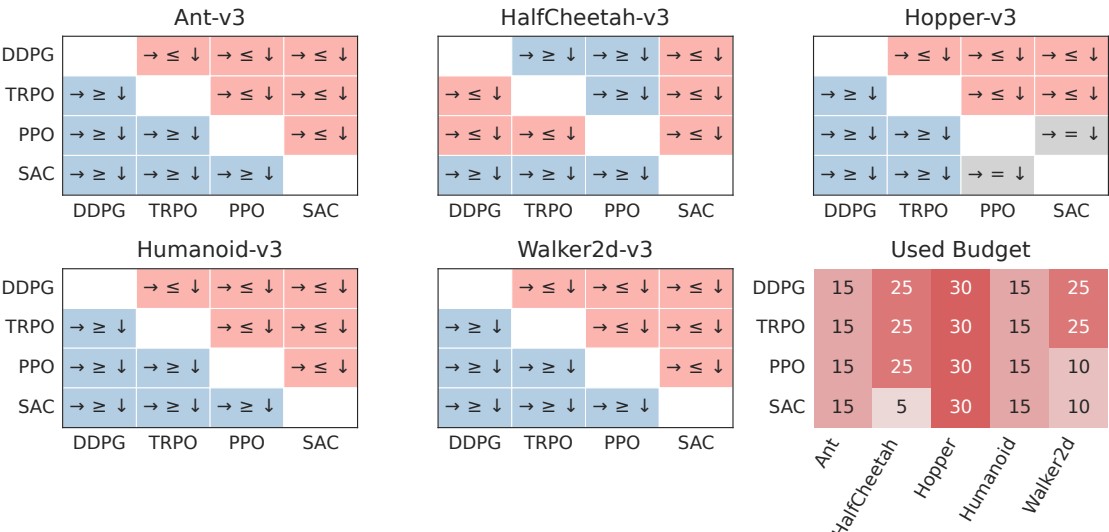

Figure 5: ADASTOP decision tables for each MuJoCo environment, and the budget used to make these decisions (bottom right). The notation →≥↓ means that the agent which name is on the left of the row is MLB than the agent which name is at the bottom of the column. See Appendix H.2 for further details.

than on-policy algorithms. Furthermore, SAC typically outperforms DDPG in continuous control robotics tasks due to its ability to handle stochastic policies, while DDPG restricts itself to deterministic policies (Haarnoja et al., 2018). Finally, PPO is generally considered performing better than TRPO in terms of cumulative reward (Engstrom et al., 2020).

For each algorithm, we fix the hyperparameters to those used by the library authors in their benchmarks for one of the MuJoCo environments. Appendix H.2 lists the values that were used and we further discuss the experimental setup. We compare the four agents in each environment using ADASTOP with $N = 5$ and $K = 6$. Fig. 5 shows the ADASTOP decision tables for each environment, as well as the number of scores per agent and environment. As expected, SAC is MLB than all the other agents in each environment. In contrast, all agents are MLB than DDPG; this may be due to the restriction to deterministic policies which hurts exploration in high-dimensional continuous control environments such as the MuJoCo benchmarks. Furthermore, we observe that the expected ordering between PPO and TRPO is generally respected, with TRPO MLB than PPO in only one environment. Finally, we note that PPO performs particularly well in some environments obtaining scores that are comparable to those of SAC, while also being the worst-performing algorithm on HalfCheetah-v3. Overall, the ADASTOP rankings in these experiments are not unexpected.

Moreover, our experiments demonstrate that ADASTOP can make decisions with fewer scores, thus reducing the computational cost of comparing Deep RL agents. For instance, as expected, SAC is MLB than all the other agents on the environment HalfCheetah-v3, and ADASTOP required only five scores to make all decisions involving SAC. Additionally, we observed that the decisions requiring the entire budget of $NK = 30$ scores were the ones in which ADASTOP determined that the agents were equivalent in terms of their scores. This decision process can be sped-up by using the early accept heuristic which is presented in the Appendix (Section F). For instance in the Walker2d-v3 environment, early accept allows us to take all the decisions after only 10 scores have been collected for each agent.

## 6 Conclusion and future works

In this paper, we introduce ADASTOP which is a sequential group test aiming at ranking agents based on their practical performance. ADASTOP may be applied in various fields where the performance of algorithms are random, such as machine learning, or optimization. Our goal is to provide statistical grounding to

define the number of times a set of agents should be run to be able to confidently rank them, up to some confidence level $\alpha$. This is the first such test, and we think this is an important contribution to computational studies in reinforcement learning and other domains. From a statistical point of view, we have been able to demonstrate the soundness of ADASTOP as a statistical test. Using ADASTOP is simple. We provide open source software to use it. Experiments demonstrate how ADASTOP may be used in practice, even in a retrospective manner using logged data: this allows one to diagnose prior studies in a statistically significant way, hopefully confirming their conclusions, possibly showing that the same conclusions could have been obtained with fewer computations.

Currently, ADASTOP considers a set of agents facing one single task. Our next step will be to extend the test to experimental settings where a set of agents are compared on a collection of tasks, such as the set of Atari games or the set of Mujoco tasks in reinforcement learning. Properly dealing with such experimental settings requires a careful statistical analysis. Moreover, additional theoretical guarantees for the early accept and the non-asymptotic control of the power, FWE, and directional error of ADASTOP for the comparison of means would also greatly improve the interpretability of the conclusions of ADASTOP. We are currently investigating these questions.

As this is illustrated in the experimental section, ADASTOP can be run on already collected scores: we do not need to run anew the agents as long as scores are available. This remark calls for an effort of the community to make their scores publicly available so that it is easy for anyone to compare one's new agent with others already proposed.

## Acknowledgments

The authors would like to thank the action editor and the reviewers for their many remarks that guided us to make this paper clearer, more legible, and to improve the rigor of the exposition. O-A.Maillard and Ph.Preux acknowledge the support of the Métropole Européenne de Lille (MEL), ANR, Inria, Université de Lille, through the AI chair Apprenf number R-PILOTE-19-004-APPRENF. R.Della Vecchia acknowledges the funding received by the CHIST-ERA Project Causal eXplainations in Reinforcement Learning – CausalXRL[7]. A.Shilova acknowledges the funding from the HPC-BigData Inria Project Lab[8]. T.Mathieu acknowledges the funding received by the SR4SG Inria exploratory action[9]. M.Centa Medeiros and H.Kohler acknowledge the funding of their Ph.D.by an ANR AI_PhD@Lille grant. This research is also partially supported by the CornelIA Hauts-de-France project. All the authors acknowledge the Scool research group for its outstanding working environment.

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

# A   Index of notations

- $N_{score}$: the total number of scores used for one agent.

- $N$: the number of scores per interim.

- $K$: the total number of interims.

- $k$: the current interim number.

- $L$: the number of agents to compare.

- $A_1, A_2, \ldots, A_L$: the agents to compare.

- $\mathbb{E}[X]$: expectation of the random variable $X$.

- $\alpha$: type I error or Family-wise error of a test.

- $\beta$: type II error of a test.

- $\mathbf{S}_n$: set of all the permutations of $\{1, \ldots, n\}$.

- $\sigma$: generic notation for a permutation, $\sigma \in \mathbf{S}_n$ for some $n \in \mathbb{N}^*$.

- id: the identity permutation, *i.e.* $\mathrm{id}(k) = k$ for all $1 \leq k \leq n$ for $\mathrm{id} \in \mathbf{S}_n$.

- $B$: the number of random permutations used to approximate the test statistic.

- $\sigma_{1:k}$: shorthand for the concatenation of permutations $\sigma_1, \ldots, \sigma_k$, *i.e.* $\sigma_{1:k}(e_{n,i}) = \sigma_i(e_{n,i})$ for all $1 \leq i \leq k$.

- $H_j$: denotes hypothesis $j$ in a multiple test, $H_j'$ denotes the alternative of hypothesis $H_j$.

- $\mathbf{c}_j$: denotes a comparison. This is a couple $(l_1, l_2)$ in $\{1, \ldots, L\}^2$.

- $j$: shorthand for denoting comparison $c_j$.

- $\mathbf{C}_0$: set of all the comparisons done in ADASTOP.

- $\mathbf{C}$: current set of undecided comparisons in ADASTOP, a subset of $\mathbf{C}_0$.

- $\mathbf{C}_k$: state of $\mathbf{C}$ at interim $k$ in ADASTOP.

- $e_i(j)$ or $e_{i,k}(j)$: score corresponding to run number $i$ when doing the test for comparison $\mathbf{c}_j$ for interim $k$.

- $T_N(\sigma)$ and $T_{N,k}^{(j)}(\sigma)$: test statistic. Defined in Equation (6) and Equation (8).

- $B_{N,k}^{(\mathbf{C})}$: boundary for test statistic $T_{N,k}^{(\mathbf{C})}$, such that if $T_{N,k}^{(\mathbf{C})}(\mathrm{id}) > B_{N,k}^{(\mathbf{C})}$, then reject the set of hypotheses associated to $\mathbf{C}$.

- $\mathbf{I}$: set of true hypotheses and $\mathbf{I}^c$ its complement.

- FWE: family-wise error, see Definition 1.

- $\mathcal{N}(\mu, \sigma^2)$: law of a Gaussian probability distribution with mean $\mu$ and variance $\sigma^2$.

- $t(\mu, \nu)$: law of a translated Student probability distribution with center of symmetry $\mu$ and $\nu$ degrees of freedom.

- $\mathcal{M}_f^{\mathcal{N}}(\mu_1, \sigma_1^2; \mu_2, \sigma_2^2)$: mixture of the two normal probability distributions $\mathcal{N}(\mu_1, \sigma_1^2)$ and $\mathcal{N}(\mu_2, \sigma_2^2)$. Each component of the mixture in weigh by $f$ and $1 - f$ respectively.

- $\mathcal{M}_f^{t}(\mu_1, \nu_1; \mu_2, \nu_2)$: mixture of two Student probability distributions $t(\mu_1, \nu_1)$ and $t(\mu_2, \nu_2)$. Each component of the mixture in weigh by $f$ and $1 - f$ respectively.

- $\mathbb{P}_{H_j, j \in \mathbf{I}}$: probability distribution when $H_j, j \in \mathbf{I}$ are true and $H_j, j \notin \mathbf{I}$ are false.

# B    Basics on hypothesis testing

To be fully understood, this paper requires the knowledge of some notions of statistics. In the hope of widening the audience of this paper, we recall notions of statistics related to hypothesis testing that are essential to understand the AdaStop test.

## B.1    Type I and type II error

In its simplest form, a statistical test is aimed at deciding whether a given collection of data $X_1, \ldots, X_N$ adheres to some hypothesis $H_0$ (called the null hypothesis), or if it is a better fit for the alternative hypothesis $H_1$. Typically, the null hypothesis states that the mean of the distribution from which the $X_i$'s are sampled is equal to some $\mu_0$: $H_0 : \mu = \mu_0$ and $H_1 : \mu \neq \mu_0$ where $\mu$ is the mean of the distribution of $X_1, \ldots, X_N$. Because $\mu$ is unknown, it has to be estimated using the data. Often this is done using the empirical mean $\widehat{\mu} = \frac{1}{N} \sum_{i=1}^{N} X_i$. $\widehat{\mu}$ is a random variable and some deviation from $\mu$ is to be expected. The theory of hypothesis tests is concerned in finding a threshold $c$ such that if $|\widehat{\mu} - \mu_0| > c$ then we say that $H_0$ is false because the deviation is greater than what was expected by the theory. A slightly more complex problem is to consider two samples $X_1, \ldots, X_N$ and $Y_1, \ldots, Y_N$ and do a two-sample test deciding whether the mean of the distribution of the $X_i$'s is equal to the mean of the distribution of the $Y_i$'s.

In both cases, the result of a test is either reject $H_0$ or to not reject $H_0$. This answer is not a ground truth: there is some probability that we make an error. However, this probability of error is often controlled and can be decomposed in type I error and type II error (often denoted $\alpha$ and $\beta$ respectively, see Table 3 and Figure 6). In words, a type I error is when we conclude that the null hypothesis is false ($H_O$ is rejected, means are different) whereas means are equal. $\alpha$ is the probability that this event occurs. A type II error is when we conclude that the null hypothesis is true ($H_0$ is accepted, means are equal) whereas means are not equal. $\beta$ is the probability that this event occurs.

|  | $H_0$ is true | $H_0$ is false |
|---|---|---|
| We accept $H_0$ | No error | Type II error $\beta$ |
| We reject $H_0$ | Type I error $\alpha$ | No error |

Table 3: Type I and type II error.

The problem is not symmetric: failing to reject the null hypothesis does not mean that the null hypothesis is true. It can be that there is not enough data to reject $H_0$. It has been shown that a test cannot simultaneously minimize $\alpha$ and $\beta$. It is customary to minimize $\beta$ for a given value of $\alpha$ (typically $\alpha$ is set to 0.05). The probability to reject $H_0$ when it is false is $1 - \beta$ and it is usually called the **power** of a test.

## B.2    Directional error

Statistical tests are often formulated as bilateral problems, which means that we test $\mu = \mu_0$ versus the bilateral interval $\mu \in (-\infty, \mu_0) \cup (\mu_0, +\infty)$. In practice (and in the case of this article), after rejecting the equality, we want to know if $\mu > \mu_0$ or if $\mu < \mu_0$. This can be done using the sign of the test statistic $\widehat{\mu} - \mu_0$. However this is not a direct consequence of the test because the test did not suppose a direction. This means that in addition to the type I and type II, we can have a type III error which is the probability to say that $\mu > \mu_0$ when in fact $\mu < \mu_0$ or that $\mu < \mu_0$ when in fact $\mu > \mu_0$.

Intuitively, if the type II error is low, then type III error (Leventhal & Huynh, 1996) tends to be even lower provided that the data distribution is sufficiently concentrated around its mean. See Figure 6 (right sub-figure) for a visual representation of the errors, here the probability that $\widehat{\mu} > \mu_0 + c$ when $\mu < \mu_0$. Type III error is often negligible, not considered in practice, and even unknown to many researchers.

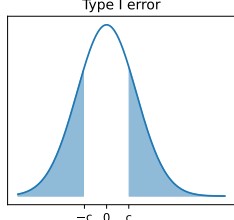 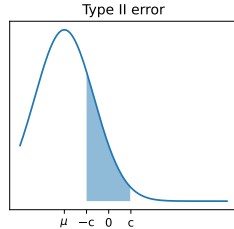 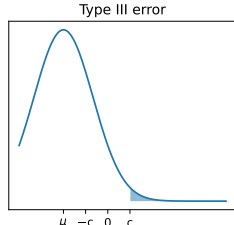

Figure 6: Illustration of type I, II and III errors: in these 3 plots, the shaded areas represent the probability of type I error (left), type II error (middle) and type III error (right) when testing $\mu = 0$ vs. $\mu \neq 0$. $c$ denotes the threshold on test statistic $\widehat{\mu}$ after which we take a decision.

### B.3 Multiple tests and FWE

When performing simultaneously $L$ statistical tests for $L > 1$, one must be careful that the error of each test accumulates. If one is not cautious, the overall error may become non-negligible. As a consequence, multiple strategies have been developed to deal with multiple testing problem.

To deal with the multiple testing problem, the first step is to define what is an error. There are several definitions of error in multiple testing among which is the False Discovery Rate which measures the expected proportion of false rejections. Another possible measure of error is the Family-Wise Error (the error we use in this article) which is defined as the probability to make at least one false rejection:

$$\text{FWE} = \mathbb{P}_{H_j, j \in \mathbf{I}} \left( \exists j \in \mathbf{I} : \quad \text{reject } H_j \right),$$

where $\mathbb{P}_{H_j, j \in \mathbf{I}}$ is used to denote a probability. $\mathbf{I} \subset \{1, \dots, L\}$ is the set of indices of the hypotheses that are true (and $\mathbf{I}^c$ the set of hypotheses that are actually false). To construct a procedure with an FWE smaller than $\alpha$, the simplest method is perhaps Bonferroni's correction (Bonferroni, 1936) in which a statistical test is applied on each of the $J$ couples of hypotheses to be tested ($J = L(L-1)/2$). And then, one would tune each hypothesis test to have a type I error $\alpha/J$. The union bound then implies that the FWE is bounded by $\alpha$:

$$\text{FWE} = \mathbb{P}_{H_j, j \in \mathbf{I}} \left( \bigcup_{j \in \mathbf{I}} \{\text{reject } H_j\} \right) \leq \sum_{i \in \mathbf{I}} \mathbb{P}_{H_j, j \in \mathbf{I}} \left( \text{reject } H_j \right) \leq |\mathbf{I}| \frac{\alpha}{J} \leq \alpha.$$

Bonferroni's correction has the advantage of being very simple to implement. However, it is often very conservative and the FWE is most often a lot smaller than $\alpha$. An alternative method that performs well in practice is the step-down method that we use in this article and is presented in Section 3.2.4.

## C Proof of Theorem 1

The proof of Theorem 1 is based on an extension of the proof of the control of FWE in the non-sequential case and the proof of the step-down method (Romano & Wolf, 2003). The interested reader may refer to Lemma 1, in the Appendix C.1, where we reproduce the proof of the bound on FWE for simple permutation tests as it is a good introduction to permutation tests. The proof proceeds as follows: first, we prove weak control[10] on the FWE by decomposing the error as the sum of the errors on each interim and using the properties of permutation tests to show that the error done at each interim is controlled by $\alpha/K$. Then, using the step-down method construction, we show that the strong control of the FWE is a consequence of the weak control because of monotony properties on the boundary values of a permutation test.

### C.1 Simplified proof for $L = 2$ agents, and $K = 1$

The proof of this theorem is a bit technical. We begin by showing it in a very simplified case where $L = 2$ agents, and $K = 1$.

---

[10]A procedure controls has a weak control on the FWE if the FWE is smaller than $\alpha$ when all null hypotheses are true. On the other hand, strong control is when FWE is smaller than $\alpha$ whatever the set of true null hypotheses.

**Lemma 1.** *Let $X_1, \ldots, X_N$ be i.i.d. from a distribution $P$ and $Y_1, \ldots, Y_N$ be i.i.d. from a distribution $Q$. Let $Z_1^{2N} = X_1, \ldots, X_N, Y_1, \ldots, Y_N$ be the concatenation of $X_1^N$ and $Y_1^N$. Let $\alpha \in (0,1)$ and define $B_N$ as:*

$$B_N = \inf \left\{ b > 0 : \quad \frac{1}{(2N)!} \sum_{\sigma \in S_{2N}} \mathbb{1} \left\{ \frac{1}{N} \sum_{i=1}^N (Z_{\sigma(i)} - Z_{\sigma(N+i)}) > b \right\} \le \alpha \right\}.$$

*Then, if $P = Q$, we have:*

$$\mathbb{P} \left( \frac{1}{N} \sum_{i=1}^N (X_i - Y_i) > B_N \right) \le \alpha.$$

*Proof.* Let us denote $T(\sigma) = \frac{1}{N} \sum_{i=1}^N (Z_{\sigma(i)} - Z_{\sigma(n+i)})$. Since $P = Q$, for any $\sigma, \sigma' \in \mathbf{S}_{2N}$ we have $T(\sigma) \overset{d}{=} T(\sigma')$. Then, because $B_N$ does not depend on the permutation $\sigma$ (but it depends on the values of $Z_1^{2N}$), we have, for any $\sigma \in \mathbf{S}_{2N}$

$$\mathbb{P}\left( T(\mathrm{id}) > B_N \right) = \mathbb{P}\left( T(\sigma) > B_N \right).$$

Now, we sum all the permutations:

$$\mathbb{P}\left( T(\mathrm{id}) > B_N \right) = \frac{1}{(2N)!} \sum_{\sigma \in \mathbf{S}_{2N}} \mathbb{E}\left[ \mathbb{1}\{ T(\sigma) > B_N \} \right]$$

$$= \mathbb{E}\left[ \frac{1}{(2N)!} \sum_{\sigma \in \mathbf{S}_{2N}} \mathbb{1}\{ T(\sigma) > B_N \} \right] \le \alpha,$$

which proves the result. $\qquad\square$

Next, we prove weak control in the general case.

### C.2 Proof of Theorem 1

In this section, we use the shorthand $\mathbb{P}$ instead of $\mathbb{P}_{H_j, j \in \mathbf{I}}$ and omit $H_{j, j \in \mathbf{I}}$ because $\mathbf{I}$ will always be the set of true hypotheses and the meaning should be clear from the context.

**Weak control on FWE:** First, we prove weak control on the FWE. This means that we suppose that all the hypotheses are true ($\mathbf{I} = \{1, \ldots, J\}$), and we control the probability to make at least one rejection. We have:

$$\mathrm{FWE} = \mathbb{P}\left( \exists j \in \mathbf{I} : \quad H_j \text{ is rejected} \right).$$

We decompose the FWE on the set of interims, using the fact that a rejection happens if and only if we reject a true hypothesis for the first time at interim $k$ for some $k = 1, \ldots, K$. These events are mutually exclusive therefore we have that their probabilities sum up and we can rewrite the FWE as:

$$\mathrm{FWE} = \sum_{k=1}^K \mathbb{P}\left( \overline{T}_{N,k}^{(\mathbf{I})}(\mathrm{id}) > B_{N,k}^{(\mathbf{I})}, \mathrm{NR}_k(\mathrm{id}) \right), \tag{13}$$

where $\mathrm{NR}_k(\mathrm{id})$ is the event on which we did *Not Reject* (NR) before interim $k$. $\mathrm{NR}_k(\mathrm{id})$ can be defined directly for a concatenation of permutations $\sigma_{1:k}$ as $\mathrm{NR}_k(\sigma_{1:k}) = \{ \forall m < k, \overline{T}_{N,m}^{(\mathbf{I})}(\sigma_{1:k}) \le B_{N,m}^{(\mathbf{I})} \}$. We introduced the definition for $\sigma_{1:k}$ and not only for id since this will be useful later on, for example in Equation (14).

Then, similarly as in the proof of Lemma 1, we want to use the invariance by permutation to make the link with the definition of $B_{N,k}^{(\mathbf{I})}$. For this purpose, we introduce the following lemma, that we prove in Appendix D.

**Lemma 2.** *For $k \le K$, for any $\sigma_{1:k}$ concatenation of $k$ permutations:*

$$(\overline{T}_{N,l}^{(\mathbf{I})}(\mathrm{id}), B_{N,l}^{(\mathbf{I})})_{l \le k} \overset{d}{=} (\overline{T}_{N,l}^{(\mathbf{I})}(\sigma_{1:l}), B_{N,l}^{(\mathbf{I})})_{l \le k}.$$

Using Lemma 2, we have for any interim $k$ and concatenation of permutations $\sigma_{1:k}$

$$\mathbb{P}\left(\overline{T}_{N,k}^{(\mathbf{I})}(\mathrm{id}) > B_{N,k}^{(\mathbf{I})}, \mathrm{NR}_k(\mathrm{id})\right) = \mathbb{P}\left(\overline{T}_{N,k}^{(\mathbf{I})}(\sigma_{1:k}) > B_{N,k}^{(\mathbf{I})}, \mathrm{NR}_k(\sigma_{1:k})\right). \tag{14}$$

Combing this equation with equation (13), we have:

$$\mathrm{FWE} \leq \sum_{k=1}^{K} \frac{1}{m_k} \sum_{\sigma_{1:k} \in \mathcal{S}_{N,k}} \mathbb{P}\left(\overline{T}_{N,k}^{(\mathbf{I})}(\sigma_{1:k}) > B_{N,k}^{(\mathbf{I})}, \mathrm{NR}_k(\sigma_{1:k})\right)$$

$$= \sum_{k=1}^{K} \mathbb{E}\left[\frac{1}{m_k} \sum_{\sigma_{1:k} \in \mathcal{S}_{N,k}} \mathbb{1}\left\{\overline{T}_{N,k}^{(\mathbf{I})}(\sigma_{1:k}) > B_{N,k}^{(\mathbf{I})}, \mathrm{NR}_k(\sigma_{1:k})\right\}\right].$$

Then, use that $\sigma_{1:k} \in \widehat{\mathcal{S}}_{N,k}$ if and only if $\sigma_{1:k} \in \mathcal{S}_{N,k}$ and $\mathrm{NR}_k(\sigma_{1:k})$ is true. Hence,

$$\mathrm{FWE} \leq \sum_{k=1}^{K} \mathbb{E}\left[\frac{1}{m_k} \sum_{\sigma_{1:k} \in \widehat{\mathcal{S}}_{N,k}} \mathbb{1}\left\{\overline{T}_{N,k}^{(\mathbf{I})}(\sigma_{1:k}) > B_{N,k}^{(\mathbf{I})}\right\}\right] \leq \sum_{k=1}^{K} q_k \leq \alpha,$$

where we use the definition of $B_{N,k}^{(\mathbf{I})}$ to make the link with $\alpha$

**Strong control of FWE:** To prove strong control, it is sufficient to show the following Lemma (see Appendix D for a proof), which is an adaptation of the proof of step-down multiple-test strong control of FWE from (Romano & Wolf, 2003).

**Lemma 3.** *Suppose that* $\mathbf{I} \subset \{1, \ldots, J\}$ *is the set of true hypotheses. We have:*

$$\mathrm{FWE} = \mathbb{P}\left(\exists j \in \mathbf{I}: H_j \text{ is rejected}\right) \leq \mathbb{P}\left(\exists k \leq K: \quad \overline{T}_{N,k}^{(\mathbf{I})}(\mathrm{id}) > B_{N,k}^{(\mathbf{I})}\right).$$

Lemma 3 shows that to control the FWE, it is sufficient to control the probability to reject on $\mathbf{I}$ given by $\mathbb{P}\left(\exists k \leq K: \quad \overline{T}_{N,k}^{(\mathbf{I})}(\sigma_{1:k}) > B_{N,k}^{(\mathbf{I})}\right)$ and this quantity, in turn, is exactly the FWE of the restricted problem of testing $(H_j)_{j \in \mathbf{I}}$ against $(H_j')_{j \in \mathbf{I}}$. In other words, Lemma 3 says that to prove strong FWE control for ADASTOP, it is sufficient to prove weak FWE control, and we already did that in the first part of the proof.

## D Proof of Lemmas 2 and 3

### D.1 Proof of Lemma 2

In this section, for an easier understanding, we change the notation for the score $e_{n,k}^{(j)}(\sigma)$ to $e_{n,k}(A_l)$ the $n^{th}$ score of agent $A_l$ at interim $k$. In other words, we change the notation of the comparison of agent $A_{l_1}$ versus agent $A_{l_2}$: in the main text a comparison was denoted by $\mathbf{c}_j \in \{1, \ldots, J\}^2$ but here we make explicit the agent from which the score has been computed resulting in the following equalities: $e_{n,k}(A_{l_1}) = e_{n,k}^{(j)}(\mathrm{id})$ and $e_{n,k}(A_{l_2}) = e_{N+n,k}^{(j)}(\mathrm{id})$ for $n \leq N$.

We denote the comparisons by $(\mathbf{c}_i)_{i \in \mathbf{I}}$. The set of comparisons can be represented as a graph in which each node represents one of the agents to compare, and there exists an edge from $j_1$ to $j_2$ denoted $(j_1, j_2)$ if $(j_1, j_2) \in (\mathbf{c}_i)_{i \in \mathbf{I}}$ is one of the comparisons that corresponds to a true hypothesis. This graph is not necessarily connected. We denote $C(i)$ the connected component to which node $l$ (*e.g.* agent $l$) belongs, *i.e.* for any $l_1, l_2 \in C(l)$ there exists a path going from $l_1$ to $l_2$. $C(l)$ cannot be equal to the singleton $\{l\}$, because this would mean that all the comparisons with $l$ are in fact false hypotheses, and then $l$ would not belong to a couple in $\mathbf{I}$.

Then, it follows from the construction of permutation test that jointly on $k \leq K$ and $\mathbf{c}_j = (l_1, l_2) \in C(l)$, we have $T_{N,k}^{(j)}(\text{id}) \stackrel{d}{=} T_{N,1}^{(j)}(\sigma_{1:k})$ for any $\sigma_1, \ldots, \sigma_k \in \mathbf{S}_{2N}$.

Let us illustrate that on an example. Suppose that $N = 2$ and $J = 3$ so that the comparison are $(A_1, A_2)$, $(A_1, A_3)$, $(A_2, A_3)$. Consider the permutation

$$\sigma_1 = \begin{pmatrix} 1 & 2 & 3 & 4 \\ 3 & 1 & 2 & 4 \end{pmatrix}$$

Because all the scores are i.i.d., we have the joint equality in distribution:

$$\begin{pmatrix} |e_{1,1}(A_1) + e_{2,1}(A_1) - e_{1,1}(A_2) - e_{2,1}(A_2)| \\ |e_{1,1}(A_3) + e_{2,1}(A_3) - e_{1,1}(A_2) - e_{2,1}(A_2)| \\ |e_{1,1}(A_1) + e_{2,1}(A_1) - e_{1,1}(A_3) - e_{2,1}(A_3)| \end{pmatrix} \stackrel{d}{=} \begin{pmatrix} |e_{2,1}(A_1) + e_{1,1}(A_2) - e_{1,1}(A_1) - e_{2,1}(A_2)| \\ |e_{2,1}(A_3) + e_{1,1}(A_2) - e_{1,1}(A_3) - e_{2,1}(A_2)| \\ |e_{2,1}(A_1) + e_{1,1}(A_3) - e_{1,1}(A_1) - e_{2,1}(A_3)| \end{pmatrix}$$

and hence,

$$(T_{N,1}^{(j)}(\text{id}))_{1 \leq j \leq 3} \stackrel{d}{=} (T_{N,1}^{(j)}(\sigma_1))_{1 \leq j \leq 3}.$$

For $k = 2$, we have for $\sigma_2 = \sigma_1$,

$$\begin{pmatrix} |e_{1,1}(A_1) + e_{2,1}(A_1) - e_{1,1}(A_2) - e_{2,1}(A_2)| \\ |e_{1,1}(A_3) + e_{2,1}(A_3) - e_{1,1}(A_2) - e_{2,1}(A_2)| \\ |e_{1,1}(A_1) + e_{2,1}(A_1) - e_{1,1}(A_3) - e_{2,1}(A_3)| \\ |e_{1,1}(A_1) + e_{2,1}(A_1) - e_{1,1}(A_2) - e_{2,1}(A_2) + e_{1,2}(A_1) + e_{2,2}(A_1) - e_{1,2}(A_2) - e_{2,2}(A_2)| \\ |e_{1,1}(A_3) + e_{2,1}(A_3) - e_{1,1}(A_2) - e_{2,1}(A_2) + e_{1,2}(A_3) + e_{2,2}(A_3) - e_{1,2}(A_2) - e_{2,2}(A_2)| \\ |e_{1,1}(A_1) + e_{2,1}(A_1) - e_{1,1}(A_3) - e_{2,1}(A_3) + e_{1,2}(A_1) + e_{2,2}(A_1) - e_{1,2}(A_3) - e_{2,2}(A_3)| \end{pmatrix}$$

$$\stackrel{d}{=} \begin{pmatrix} |e_{1,1}(A_1) + e_{2,1}(A_2) - e_{1,1}(A_1) - e_{2,1}(A_2)| \\ |e_{1,1}(A_3) + e_{2,1}(A_2) - e_{1,1}(A_3) - e_{2,1}(A_2)| \\ |e_{1,1}(A_1) + e_{2,1}(A_3) - e_{1,1}(A_1) - e_{2,1}(A_3)| \\ |e_{1,1}(A_1) + e_{2,1}(A_2) - e_{1,1}(A_1) - e_{2,1}(A_2) + e_{1,2}(A_1) + e_{2,2}(A_2) - e_{1,2}(A_1) - e_{2,2}(A_2)| \\ |e_{1,1}(A_3) + e_{2,1}(A_2) - e_{1,1}(A_3) - e_{2,1}(A_2) + e_{1,2}(A_3) + e_{2,2}(A_2) - e_{1,2}(A_3) - e_{2,2}(A_2)| \\ |e_{1,1}(A_1) + e_{2,1}(A_3) - e_{1,1}(A_1) - e_{2,1}(A_3) + e_{1,2}(A_1) + e_{2,2}(A_3) - e_{1,2}(A_1) - e_{2,2}(A_3)| \end{pmatrix}$$

and then, we get jointly

$$(T_{N,k}^{(j)}(\text{id}))_{1 \leq j \leq 3, k \leq 2} \stackrel{d}{=} (T_{N,k}^{(j)}(\sigma_1 \cdot \sigma_2))_{1 \leq j \leq 3, k \leq 2}.$$

This reasoning can be generalized to any $N$, $J$ and $K$:

$$(T_{N,k}^{(j)}(\text{id}))_{k \leq K, \mathbf{c}_j \in C(l)^2} \stackrel{d}{=} (T_{N,k}^{(j)}(\sigma_{1:k}))_{k \leq K, \mathbf{c}_j \in C(l)^2}.$$

Then, by construction, the different connected component $C(l)$ are independent of one another and hence,

$$(T_{N,k}^{(j)}(\text{id}))_{k \leq K, j \in \mathbf{I}} \stackrel{d}{=} (T_{N,k}^{(j)}(\sigma_{1:k}))_{k \leq K, j \in \mathbf{I}}.$$

Lemma 2 follows from taking the maximum on all the comparisons and because the boundaries do not depend on the permutation.

## D.2 Proof of Lemma 3

Denote by $\mathbf{C}_k$ the (random) value of $\mathbf{C}$ at the beginning of interim $k$. We have:

$$\text{FWE} = \mathbb{P}\left(\exists j \in \mathbf{I}: \quad H_j \text{ is rejected}\right)$$

$$= \mathbb{P}\left(\exists k \leq K: \quad \overline{T}_{N,k}^{(\mathbf{C}_k)}(\text{id}) > B_{N,k}^{(\mathbf{C}_k)}, \underset{j, \mathbf{c}_j \in \mathbf{C}_k}{\arg\max} \, \overline{T}_{N,k}^{(j)}(\text{id}) \in \mathbf{I}\right). \tag{15}$$

Then, let $k_0$ correspond to the very first rejection (if any) in the algorithm. Having that the argmax is attained in $\mathbf{I}$,

$$\overline{T}_{N,k_0}^{(\mathbf{C}_{k_0})}(\mathrm{id}) = \max\{T_{N,k_0}^{(j)}(\mathrm{id}), \mathbf{c}_j \in \mathbf{C}_{k_0}\} = \max\{T_{N,k_0}^{(j)}(\mathrm{id}), j \in \mathbf{I}\} = \overline{T}_{N,k_0}^{(\mathbf{I})}(\mathrm{id})$$

Moreover, having that the comparisons indexed by $\mathbf{I}$ are included into $\mathbf{C}_{k_0}$, we have $B_{N,k_0}^{(\mathbf{C}_{k_0})} \geq B_{N,k_0}^{(\mathbf{I})}$. Injecting these two relations in Equation (15), we obtain:

$$\mathrm{FWE} \leq \mathbb{P}\left(\exists k \leq K: \quad \overline{T}_{N,k}^{(\mathbf{I})}(\mathrm{id}) > B_{N,k}^{(\mathbf{I})}, \underset{j \in \mathbf{C}_k}{\arg\max}\, \overline{T}_{N,k}^{(j)}(\mathrm{id}) \in \mathbf{I}\right)$$

$$\leq \mathbb{P}\left(\exists k \leq K: \quad \overline{T}_{N,k}^{(\mathbf{I})}(\mathrm{id}) > B_{N,k}^{(\mathbf{I})}\right).$$

This proves the desired result.

# E    Asymptotic results for two agents

## E.1    Convergence of boundaries and comparing the means

Because there are only two agents and no early stopping, we simplify the notations and denote

$$t_{N,i}(\sigma_i) = \sum_{n=1}^{N} e_{\sigma_i(n),i}(2) - \sum_{n=N+1}^{2N} e_{\sigma_i(n),i}(1),$$

and

$$T_{N,k}(\sigma_{1:k}) = \left| \sum_{i=1}^{k} \left( \sum_{n=1}^{N} e_{\sigma_i(n),i}(2) - \sum_{n=N+1}^{2N} e_{\sigma_i(n),i}(1) \right) \right|$$

$$= \left| \sum_{i=1}^{k} t_{N,i}(\sigma_i) \right|,$$

and

$$B_{N,k} = \inf\left\{ b > 0 : \frac{1}{((2N)!)^k} \sum_{\sigma_1,\dots,\sigma_k \in \mathbf{S}_{2N}^k} \mathbb{1}\{T_{N,k}(\sigma_{1:k}) \geq b\} \leq q_k \right\}.$$

When there is only one interim $(K = 1)$, we have the following convergence of the randomization law of $T_{N,1}(\sigma)$.

**Proposition 1** (Theorem 17.3.1 in (Lehmann et al., 2005)). *Suppose $e_{1,1}(1),\dots,e_{N,1}(1)$ are i.i.d.from $P$ and $e_{1,1}(2),\dots,e_{N,1}(2)$ are i.i.d.from $Q$ and both $P$ and $Q$ have finite variance. Then, we have:*

$$\sup_t \left| \frac{1}{(2N)!} \sum_{\sigma \in \mathbf{S}_{2N}} \mathbb{1}\left\{ \frac{1}{\sqrt{N}} T_{N,1}(\sigma) \leq t \right\} - \Phi\left(t/\tau(P,Q)\right) \right| \xrightarrow[N\to\infty]{P} 0$$

*where $\Phi$ is the normal c.d.f.and $\tau(P,Q)^2 = \sigma_P^2 + \sigma_Q^2 + \frac{(\mu_P - \mu_Q)^2}{2}$.*

Using the non-sequential result from proposition 1, we can show the following theorem that controls the asymptotic law of the sequential test.

**Theorem 3.** *Suppose that $P$ and $Q$ both have a finite second moment. Then, for any $1 \leq k \leq K$, $\frac{1}{\sqrt{N}} B_{N,k} \xrightarrow[N\to\infty]{} b_k$ where the real numbers $b_k$ are defined as follows. Let $W_1,\dots,W_K$ be i.i.d.random variables with law $\mathcal{N}(0,1)$. Then $b_1$ is the solution of the following equation:*

$$\mathbb{P}\left(|W_1| \geq \frac{b_1}{\tau(P,Q)}\right) = \frac{\alpha}{K},$$

*and for any $1 < k \le K$, $b_k$ is the solution of*

$$\mathbb{P}\left(\left|\frac{1}{k}\sum_{j=1}^{k}W_j\right| > \frac{b_l}{\tau(P,Q)}, \quad \forall j < k, \left|\frac{1}{j}\sum_{i=1}^{j}W_i\right| \le \frac{b_j}{\tau(P,Q)}\right) = \frac{\alpha}{K}.$$

The conditions on Theorem 3 are very weak: the existence of a finite second moment to be able to use the central limit theorem.

The test performed in ADASTOP when comparing two agents corresponds to testing

$$\mathbb{1}\left\{\exists k \le K : \frac{1}{\sqrt{N}}T_{N,k}(\mathrm{id}) > \frac{1}{\sqrt{N}}B_{N,k}\right\}$$

and from Theorem 3 and the central-limit theorem, $\frac{1}{\sqrt{N}}T_{N,k}(\mathrm{id})$ converges to $\sum_{j=1}^{k}W_j\sqrt{\sigma_P^2 + \sigma_Q^2}$, and $B_{N,k}/\sqrt{N}$ converges to $b_k$. Hence the test is asymptotically equivalent to:

$$\mathbb{1}\left\{\exists k \le K : \sum_{j=1}^{k}W_j\sqrt{\sigma_P^2 + \sigma_Q^2} > b_k\right\}.$$

Then, in the case in which $\mu_P = \mu_Q$, we have $\tau(P,Q) = \sqrt{\sigma_P^2 + \sigma_Q^2}$ and

$$\begin{aligned}
\mathrm{FWE} &= \mathbb{P}\left(\exists k \le K : \sum_{j=1}^{k}W_j\sqrt{\sigma_P^2 + \sigma_Q^2} > b_k\right)\\
&= \sum_{k=1}^{K}\mathbb{P}\left(\left|\frac{1}{k}\sum_{j=1}^{k}W_j\right| > \frac{b_l}{\tau(P,Q)}, \quad \forall j < k, \left|\frac{1}{j}\sum_{i=1}^{j}W_i\right| \le \frac{b_j}{\tau(P,Q)}\right)\\
&= \sum_{k=1}^{K}\frac{\alpha}{K} = \alpha.
\end{aligned}$$

Hence, for the test $H_0: \quad \mu_P = \mu_Q$ versus $H_1: \quad \mu_P \ne \mu_Q$, our test has asymptotic type I error $\alpha$.

### E.2 Proof of Theorem 3

For $x \in \mathbb{R}$, we denote:

$$R_{N,k}(x) = \frac{1}{(2N)!}\sum_{\sigma_k \in \mathbf{S}_{2N}}\mathbb{1}\{t_{N,k}(\sigma_k) \le x\}.$$

$R_{N,k}$ is the c.d.f.of the randomization law of $t_{N,k}(\sigma_k)$, and by Proposition 1, it converges uniformly to a Gaussian c.d.f when $N$ goes to infinity.

**Convergence of $B_{N,1}$** By definition of the boundary, we have

$$\frac{1}{\sqrt{N}}B_{N,1} = \frac{1}{\sqrt{N}}\min\left\{b > 0 : \frac{1}{(2N)!}\sum_{\sigma_1 \in \mathbf{S}_{2N}}\mathbb{1}\{|T_{N,1}(\sigma_1)| > b\} \le \frac{\alpha}{K}\right\}.$$

This implies

$$\frac{1}{(2N)!}\sum_{\sigma_1 \in \mathbf{S}_{2N}}\mathbb{1}\{|T_{N,1}(\sigma_1)| \le B_{N,1}\} = \widehat{R}_{N,1}\left(\frac{1}{\sqrt{N}}B_{N,1}\right) - \widehat{R}_{N,1}\left(-\frac{1}{\sqrt{N}}B_{N,1}\right)$$

$$\ge 1 - \frac{\alpha}{K}$$

and for any $b < B_{N,1}$, we have:

$$\frac{1}{(2N)!} \sum_{\sigma_1 \in \mathbf{S}_{2N}} \mathbb{1}\{|T_{N,1}(\sigma_1)| \le b\} = \widehat{R}_{N,1}\left(\frac{b}{\sqrt{N}}\right) - \widehat{R}_{N,1}\left(-\frac{b}{\sqrt{N}}\right) < 1 - \frac{\alpha}{K}.$$

Then,

$$\Phi\left(\frac{B_{N,1}}{\tau(P,Q)\sqrt{N}}\right) - \Phi\left(-\frac{B_{N,1}}{\tau(P,Q)\sqrt{N}}\right)$$

$$\ge \widehat{R}_{N,1}\left(\frac{B_{N,1}}{\sqrt{N}}\right) - \widehat{R}_{N,1}\left(-\frac{B_{N,1}}{\sqrt{N}}\right) - \left|\Phi\left(\frac{B_{N,1}}{\tau(P,Q)\sqrt{N}}\right) - \widehat{R}_{N,1}\left(\frac{B_{N,1}}{\sqrt{N}}\right)\right|$$

$$- \left|\Phi\left(-\frac{B_{N,1}}{\tau(P,Q)\sqrt{N}}\right) - \widehat{R}_{N,1}\left(-\frac{B_{N,1}}{\sqrt{N}}\right)\right|$$

$$\ge 1 - \frac{\alpha}{K} - 2\sup_t \left|\Phi\left(\frac{t}{\tau(P,Q)}\right) - \widehat{R}_{N,1}(t)\right|.$$

Hence, by taking $N$ to infinity, we have from Proposition 1:

$$\liminf_{N \to \infty} \Phi\left(\frac{B_{N,1}}{\tau(P,Q)\sqrt{N}}\right) - \Phi\left(-\frac{B_{N,1}}{\tau(P,Q)\sqrt{N}}\right) \ge 1 - \frac{\alpha}{K}.$$

And for any $\varepsilon > 0$, because of the definition of $B_{N,1}$ as a supremum, we have:

$$\limsup_{N \to \infty} \Phi\left(\frac{B_{N,1} + \varepsilon}{\tau(P,Q)\sqrt{N}}\right) - \Phi\left(-\frac{B_{N,1} + \varepsilon}{\tau(P,Q)\sqrt{N}}\right) < 1 - \frac{\alpha}{K}.$$

By continuity of $\Phi$, this implies that $\frac{1}{\sqrt{N}}B_{N,1}$ converges almost surely and its limit is such that:

$$\Phi\left(\frac{\lim_{N \to \infty} B_{N,1}/\sqrt{N}}{\tau(P,Q)}\right) - \Phi\left(-\frac{\lim_{N \to \infty} B_{N,1}/\sqrt{N}}{\tau(P,Q)}\right) = 1 - \frac{\alpha}{K}.$$

Or said differently, let $W \sim \mathcal{N}(0,1)$, then we have the almost sure convergence $\lim_{N \to \infty} \frac{1}{\sqrt{N}}B_{N,1} = b_1$ where $b_1$ is the real number defined by

$$\mathbb{P}\left(|W| \ge \frac{b_1}{\tau(P,Q)}\right) = \frac{\alpha}{K}.$$

**Convergence of $B_{N,k}$ for $k > 1$.** We proceed by induction. Suppose that $\frac{1}{\sqrt{N}}B_{N,k-1}$ converges to some $b_{k-1} > 0$ and that for any $d_1, \ldots, d_{k-1}$, the randomization probability, it holds that:

$$\sup_{d_1, \ldots, d_{k-1}} \left| \frac{1}{((2N)!)^{k-1}} \sum_{\sigma_1, \ldots, \sigma_{k-1} \in \mathbf{S}_{2N}} \mathbb{1}\left\{\forall j \le k-1, \sum_{i=1}^{j} t_{N,i}(\sigma_i) \le d_j\sqrt{N}\right\} \right.$$

$$\left. - \mathbb{P}\left(\forall j \le k-1, \sum_{i=1}^{j} W_i \le \frac{d_j}{\tau(P,Q)}\right) \right| \xrightarrow[a.s.]{N \to \infty} 0, \quad (16)$$

where $W_1, \ldots, W_{k-1}$ are i.i.d. $\mathcal{N}(0,1)$ random variables. In other words, the randomization law converges uniformly to the joint law described above with the sum of Gaussian random variables.

Then, by uniform convergence and by convergence of the $B_{N,j}$, we have that:

$$\frac{1}{((2N)!)^{k-1}} \sum_{\sigma_1, \ldots, \sigma_{k-1} \in \mathbf{S}_{2N}} \mathbb{1}\left\{T_{N,j}(\sigma_{1:j}) > B_{N,k-1}, \quad \forall j < k-1, T_{N,j}(\sigma_{1:j}) \le B_{N,j}\right\}$$

converges to

$$\mathbb{P}\left(\left|\sum_{i=1}^{l} W_i\right| > \frac{b_l}{\tau(P,Q)}, \quad \forall j < l, \left|\sum_{i=1}^{j} W_i\right| \le \frac{b_j}{\tau(P,Q)}\right) = \frac{\alpha}{K}. \tag{17}$$

the last equality follows by construction of $B_{N,j}$ for $j < k$.

We have:

$$B_{N,k} = \min\left\{ b > 0 : \frac{1}{((2N)!^k)} \sum_{\sigma_1,\ldots,\sigma_k \in \mathbf{S}_{2N}} \mathbb{1}\left\{\begin{array}{l}\left|\sum_{j=0}^{k} t_{N,j}(\sigma_j)\right| \ge b, \\ \forall j < k, \left|\sum_{i=0}^{j} t_{N,i}(\sigma_i)\right| \le B_{N,j}\end{array}\right\} + \sum_{i=1}^{k-1} q_i \le \frac{k\alpha}{K}\right\}.$$

By the induction hypothesis, we have $q_i \xrightarrow[n\to\infty]{} \alpha/K$ for any $i < k$.

Let $W_1, \ldots, W_k$ be i.i.d. $\mathcal{N}(0,1)$ random variables. We show the following lemma that proves part of the step $k$ of the induction hypothesis, proved in Section E.3.

**Lemma 4.** *Suppose Equation* (17) *is true. Then,*

$$\sup_{d_1,\ldots,d_k}\left|\frac{1}{((2N)!)^k} \sum_{\sigma_1,\ldots,\sigma_k \in \mathbf{S}_{2N}} \mathbb{1}\left\{\forall j \le k, \sum_{i=1}^{j} t_{N,i}(\sigma_i) \le d_j\sqrt{N}\right\}\right.$$

$$\left. - \mathbb{P}\left(\forall j \le k, \sum_{i=1}^{j} W_i \le \frac{d_j}{\tau(P,Q)}\right)\right| \xrightarrow[a.s.]{N\to\infty} 0,$$

Then, what remains is to prove the convergence of $B_{N,k}$. Let us denote:

$$\Psi_k(d_k) = \mathbb{P}\left(\left|\sum_{i=1}^{k} W_i\right| > \frac{d_k}{\tau(P,Q)}, \forall j \le k-1, \left|\sum_{i=1}^{j} W_i\right| \le \frac{b_j}{\tau(P,Q)}\right).$$

From Lemma 4, we have that

$$\left|\Psi_k\left(\frac{B_{N,k}}{\sqrt{N}}\right) - \frac{1}{((2N)!)^k} \sum_{\sigma_1,\ldots,\sigma_k \in \mathbf{S}_{2N}} \mathbb{1}\left\{T_{N,k}(\sigma_{1:k}) > B_{N,k}, \quad \forall j < k, T_{N,j}(\sigma_{1:j}) \le B_{N,j}\right\}\right|$$

converges to 0 as $N$ goes to infinity. Hence,

$$\limsup_{N\to\infty} \Psi_k\left(\frac{B_{N,k}}{\sqrt{N}}\right) \le \alpha/K.$$

Then, similarly to the case $k = 1$, we also have for any $\varepsilon > 0$:

$$\liminf_{N\to\infty} \Psi_k\left(\frac{B_{N,k} - \varepsilon}{\sqrt{N}}\right) \ge \alpha/K$$

and by continuity of $\Psi_k$ (which is a consequence of the continuity of the joint c.d.f. of Gaussian random variables) we conclude that $B_{N,k}/\sqrt{N}$ converges almost surely to $b_k$.

### E.3 Proof of Lemma 4

In this proof, let $E_{\sigma_{1:k}}(x)$ denote the expectation of the randomization law defined for some function $f : \mathbf{S}_{2N}^k \to \mathbb{R}$ by:

$$E_{\sigma_{1:k}}[f(\sigma_{1:k})] = \frac{1}{((2N)!)^k} \sum_{\sigma_1,\ldots,\sigma_k \in \mathbf{S}_{2N}} f(\sigma_{1:k}).$$

Please note that this expectation is still random and should be differentiated from the usual expectation $\mathbb{E}$. First, let us first handle the convergence of step $k$. We have:

$$\frac{1}{(2N)!} \sum_{\sigma_k \in \mathbf{S}_{2N}} \mathbb{1}\left\{ \sum_{j=1}^{k} t_{N,j}(\sigma_j) \leq d_k \sqrt{N} \right\}$$

$$= \frac{1}{(2N)!} \sum_{\sigma_k \in \mathbf{S}_{2N}} \mathbb{1}\left\{ \frac{1}{\sqrt{N}} t_{N,k}(\sigma_k) \leq d_k - \frac{1}{\sqrt{N}} \sum_{j=1}^{k-1} t_{N,j}(\sigma_j) \right\}$$

$$= \widehat{R}_{n,k}\left( d_k - \frac{1}{\sqrt{N}} \sum_{j=1}^{k-1} t_{N,j}(\sigma_j) \right).$$

Because the convergence in proposition 1 is uniform, we have:

$$\left| \widehat{R}_{n,k}\left( d_k - \frac{1}{\sqrt{N}} \sum_{j=1}^{k-1} t_{N,j}(\sigma_j) \right) - \Phi\left( \frac{1}{\tau(P,Q)} \left( d_k - \frac{1}{\sqrt{N}} \sum_{j=1}^{k-1} t_{N,j}(\sigma_j) \right) \right) \right|$$

$$\leq \sup_{t} \left| \widehat{R}_n(t) - \Phi\left( \frac{t}{\tau(P,Q)} \right) \right| \xrightarrow[n \to \infty]{} 0.$$

Then, using this convergence we have that, when $N$ goes to infinity,

$$E_{\sigma_{1:k}}\left[ \mathbb{1}\left\{ \forall j < k, \sum_{i=1}^{j} t_{N,i}(\sigma_i) \leq d_j \sqrt{N} \right\} \right]$$

converges uniformly on $d_1, \ldots, d_k$ to:

$$E_{\sigma_{1:k}}\left[ \mathbb{1}\left\{ \forall j < k-1, \sum_{i=1}^{j} t_{N,i}(\sigma_i) \leq d_j \sqrt{N} \right\} \mathbb{P}\left( W_k \leq \frac{1}{\tau(P,Q)} \left( d_k - \frac{1}{\sqrt{N}} \sum_{i=1}^{k-1} t_{N,i}(\sigma_i) \right) \right) \right]$$

$$= \mathbb{E}\left[ E_{\sigma_{1:k-1}}\left[ \mathbb{1}\left\{ \begin{array}{c} \forall j < k-2, \sum_{i=1}^{j} t_{N,i}(\sigma_i) \leq d_j \sqrt{N}, \\ \frac{1}{\sqrt{N}} \sum_{i=1}^{k-1} t_{N,j}(\sigma_i) \leq \min(d_k - \tau(P,Q) W_k, d_{k-1}) \end{array} \right\} \right] \right]. \tag{18}$$

Then, using the induction hypothesis, Equation (18) converges to Equation (16), hence in the limit we have

$$\mathbb{E}\left[ \mathbb{1}\left\{ \begin{array}{c} \forall j < k-2, \sum_{i=1}^{j} W_i \leq \frac{d_j}{\tau(P,Q)}, \\ \frac{1}{\sqrt{N}} \sum_{i=1}^{k-1} W_i \leq \frac{1}{\tau(P,Q)} \min(d_k - W_k, d_{k-1}) \end{array} \right\} \right] = \mathbb{E}\left[ \mathbb{1}\left\{ \forall j < k, \sum_{i=1}^{j} W_i \leq \frac{d_j}{\tau(P,Q)} \right\} \right]$$

$$= \mathbb{P}\left( \forall j \leq k, \sum_{i=1}^{j} W_i \leq \frac{d_j}{\tau(P,Q)} \right).$$

## F On early accept in AdaStop

Let $\mathbf{C} \subset \{\mathbf{c}_1, \ldots, \mathbf{c}_J\}$ be a subset of the set of comparisons that we want to do. Let us denote:

$$\overline{T}_{N,k}^{(\mathbf{C})}(\sigma_1^k) := \max\left( T_{N,k}^{(j)}(\sigma_1^k), \quad \mathbf{c}_j \in \mathbf{C} \right) \quad \text{and} \quad \underline{T}_{N,k}^{(\mathbf{C})}(\sigma_1^k) := \min\left( T_{N,k}^{(j)}(\sigma_1^k), \quad \mathbf{c}_j \in \mathbf{C} \right),$$

and

$$\overline{B}_{N,k}^{(\mathbf{C})} := \inf\left\{ b > 0 : \frac{1}{m_k} \sum_{\sigma \in \widehat{\mathcal{S}}_{N,k}} \mathbb{1}\{\overline{T}_{N,k}^{(\mathbf{C})}(\sigma_1^k) \geq b\} \leq \overline{q}_k \right\}, \tag{19}$$

and

$$\underline{B}_{N,k}^{(\mathbf{C})} := \sup\left\{ b > 0 : \frac{1}{m_k} \sum_{\sigma \in \widehat{\mathcal{S}}_{N,k}} \mathbb{1}\{\underline{T}_{N,k}^{(\mathbf{C})}(\sigma_1^k) \le b\} \le \underline{q}_k \right\} \tag{20}$$

where the $\underline{q_i} = \lfloor \frac{\alpha i}{2K}\binom{2N}{N}\rfloor/(\frac{1}{2}\binom{2N}{N}) - \underline{q}_{i-1}$ and $\overline{q}_i = \lfloor \frac{\beta i}{2K}\binom{2N}{N}\rfloor/(\frac{1}{2}\binom{2N}{N}) - \overline{q}_{i-1}$ and where $\widehat{\mathcal{S}}_{N,k}$ is defined in Equation (11). By construction of $\widehat{\mathcal{S}}_{N,k}$, for each $\sigma_1^k \in \widehat{\mathcal{S}}_{N,k}$, we have the following property

$$\forall m < k, \quad \overline{T}_{N,m}^{(\mathbf{C})}(\sigma_1^k) \le \overline{B}_{N,m}^{(\mathbf{C})} \text{ and } \underline{T}_{N,m}^{(\mathbf{C})}(\sigma_1^k) \ge \underline{B}_{N,m}^{(\mathbf{C})}.$$

In ADASTOP, modify the decision step (lines 9 to 13 in Algorithm 3) to Algorithm 4.

---

**Algorithm 4:** Early accept.

1 **if** $\overline{T}_{N,k}^{(C)}(\text{id}) > \overline{B}_{N,k}^{(C)}$ **then**

2 $\quad$ Reject $H_{j_{\max}}$ where $\mathbf{c}_{j_{\max}} = \arg\max\left(T_{N,k}^{(j)}(\text{id}), \quad \mathbf{c}_j \in \mathbf{C}\right)$.

3 $\quad$ Update $\mathbf{C} = \mathbf{C} \setminus \{\mathbf{c}_{j_{\max}}\}$

4 **else if** $\underline{T}_{n,k}^{(C)}(\text{id}) < \underline{B}_{N,k}^{(C)}$ **then**

5 $\quad$ Accept $H_{j_{\min}}$ where $\mathbf{c}_{j_{\min}} = \arg\min\left(T_{N,k}^{(j)}(\text{id}), \quad \mathbf{c}_j \in \mathbf{C}\right)$.

6 $\quad$ Update $\mathbf{C} = \mathbf{C} \setminus \{\mathbf{c}_{j_{\min}}\}$

---

The resulting algorithm has a small probability to accept a decision early, and as a consequence it may be unnecessary to compute the score of some of the agents in the subsequent steps.

As an illustration of the performance of early accept, if we run ADASTOP with early parameter $\beta = 0.01$ (used to define the boundary in Equation (20)) on the Walker2D-v3 experiment from Section 5.3, the experiment stops at interim 2 and 10 scores have been collected for each agent. This has to be compared with the fact that, in Section 5.3, we showed that without early accept, ADASTOP uses 30 scores for DDPG and TRPO. In this example, early stopping saves a lot of computations and results in a significant speed-up without affecting the final decisions.

## G  Agent comparison on Atari environments

In this section, we discuss the approach of (Agarwal et al., 2021) on the problem of comparing RL agents on Atari environments. The methodology from (Agarwal et al., 2021) prescribes to give confidence intervals around some measure of location (typically mean or interquartile mean, IQM) of the agents score. We demonstrate this approach in the left and middle sub-figures of Figure 7. However, it is not clear how to draw a conclusion from such graphs (see (Cumming & Finch, 2005) for a discussion on doing inference with confidence intervals). Moreover, there is no definite criterion on how to construct the confidence interval. For example, this approach gives rise to three very different confidence interval plots in Figure 7: the plots on the left and in the middle are the ones presented in (Agarwal et al., 2021) and the plot on the right is a modified version of the plot in the middle, where we added the error due to performing multiple comparisons.

In the leftmost plot of Figure 7, it is easy to draw conclusions but on the other hand the theoretical interpretation is not clear because the fact that different games have different laws for their scores is not taken into account. The middle and rightmost plots in Figure 7 are a naive approach to the problem that assumes that all the games are very different from one another, and as such the confidence intervals are too large to draw conclusions. We think that there should be an intermediate approach that considers a cluster of similar games (*e.g.* all the easy games, all the maze-like games, all the difficult games, etc.) and treat these games as having all the same law. This approach would produce smaller (and more interpretable) confidence intervals compared to the naive approach, providing a middle-ground between the first plot and second plot.

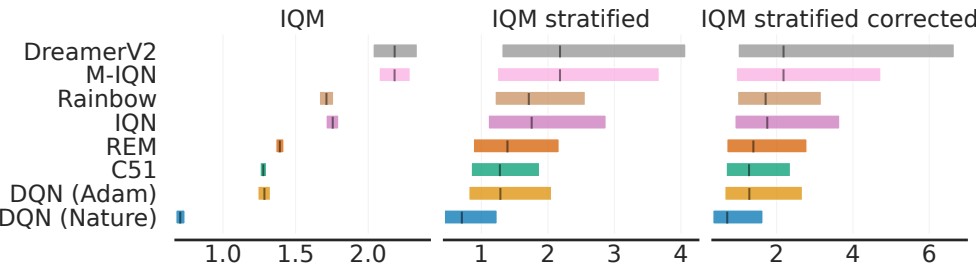

Figure 7: 95% confidence intervals on the IQM of the "Human Normalized Score" of a set of algorithms with various methods using the results from (Agarwal et al., 2021).

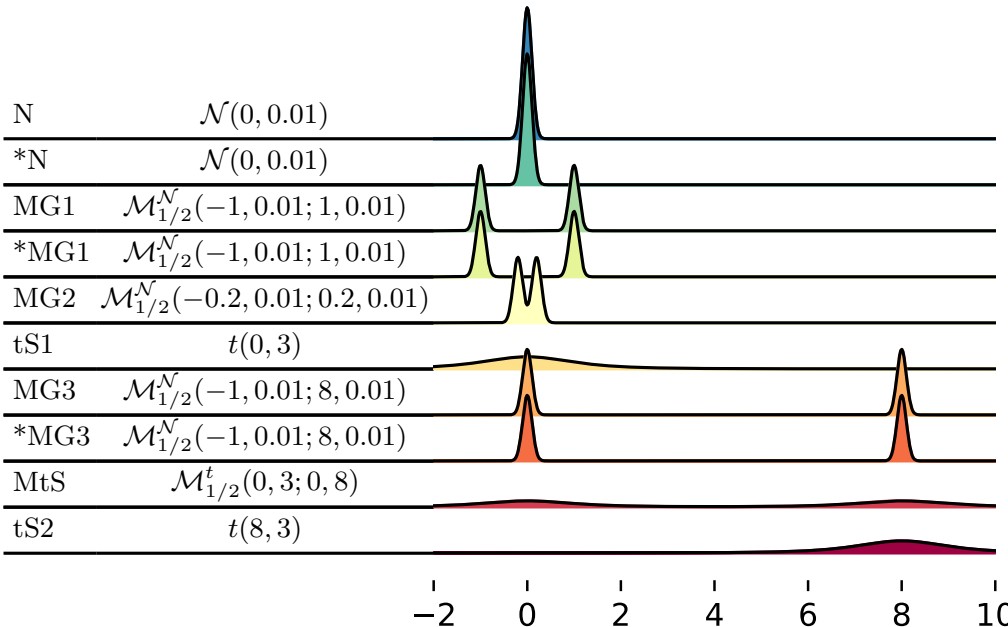

Figure 8: Toy example 3, with an illustration of the involved distributions.

## H Implementation details and additional experiments

### H.1 Complementary experiment for Section 5.1

We consider 10 agents which score distributions are listed in Fig. 8, where the first column indicates the labels of the agents as they are used in Fig. 9. We add some more notations: $t(\mu, \nu)$ is the t-Student distribution with mean $\mu$ and $\nu$ degrees of freedom, and $\mathcal{M}_1^t(t(\mu_1, \nu_1), t(\mu_2, \nu_2))$ is a mixture of 2 t-Student distributions $t(\mu_1, \nu_1)$ and $t(\mu_2, \nu_2)$, each weights $f$ and $1 - f$.

Similarly to the setup of cases 1 and 2 (see Section 5.1), we execute ADASTOP with $K = 5$, $N = 5$, $\alpha = 0.05$. As indicated in Section 4, we do not enumerate all the permutations in the permutation test as this would be too expensive. Instead we use $10^4$ randomly selected permutations to compute our test statistics at each interim.

In contrast to the experiments reported in the main text, we use early accept (with $\beta = 0.01$) to avoid situations where all agents are compared with the maximum number of scores, *i.e.* $NK$ scores. This may occur when each agent has a similar distribution to at least one other agent.

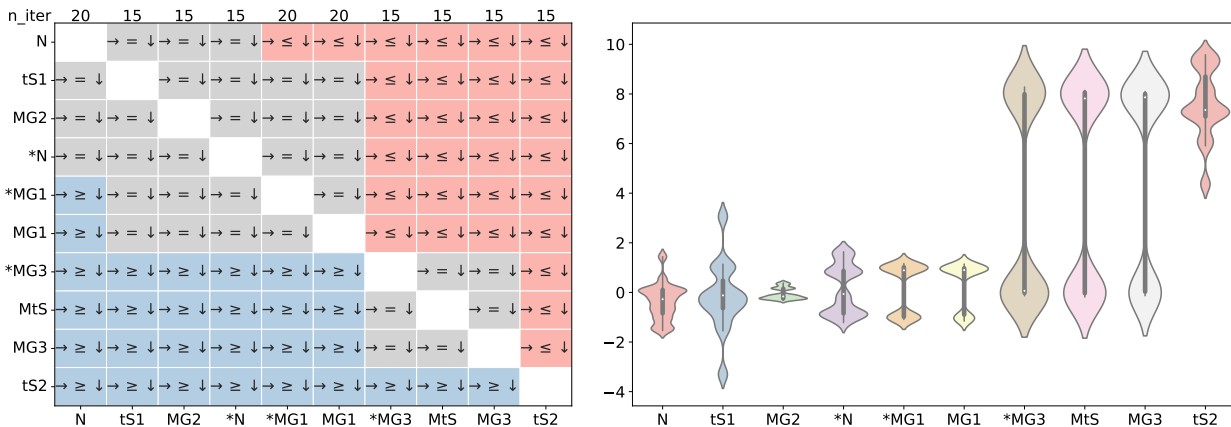

Figure 9: ADASTOP decision table (left) and measured empirical distributions (right).

We show the performance of ADASTOP for multiple agents comparison in Fig. 9, which corresponds to the output of one execution of ADASTOP. The table (left part of Fig. 9) summarizes the decisions of the algorithm for every pair of comparisons. The violin plots (right part of Fig. 9) reflect empirically measured distributions in the comparison. From this figure, we can see that almost all agents are clustered according to the mean of this distribution, except for *MG3, Mt5 and MG3 that are assigned to two different groups at once. Interestingly, except for *MG3, these clusters are correctly formed. Moreover, similarly to the two previous cases, we have executed ADASTOP $M = 5\,000$ times to measure the FWE of the test. Empirically, we measured a FWE of 0.0178 for the test comparing the distributions and a FWE of 0.0472 for the test comparing the means: both are below 0.05. This example illustrates the fact that ADASTOP can be efficiently used to compare the score of several agents simultaneously.

## H.2   MuJoCo Experiments

|  | DDPG | TRPO | PPO | SAC |
|---|---|---|---|---|
| $\gamma$ | 0.99 | 0.99 | 0.99 | 0.99 |
| Learning Rate | $1 \times 10^{-3}$ | $1 \times 10^{-3}$ | $3 \times 10^{-4}$ | $3 \times 10^{-4}$ |
| Batch Size | 128 | 64 | 64 | 256 |
| Buffer Size | $10^6$ | 1024 | 2048 | $10^6$ |
| Value Loss | MSE | MSE | AVEC (Flet-Berliac et al., 2021) | MSE |
| Use gSDE | No | No | No | Yes |
| Entropy Coef. | - | 0 | 0 | `auto` |
| GAE $\lambda$ | - | 0.95 | 0.95 | - |
| Advantage Norm. | - | Yes | Yes | - |
| Target Smoothing | 0.005 | - | - | 0.005 |
| Learning Starts | $10^4$ | - | - | $10^4$ |
| Policy Frequency | 32 | - | - | - |
| Exploration Noise | 0.1 | - | - | - |
| Noise Clip | 0.5 | - | - | - |
| Max KL | - | $10^{-2}$ | - | - |
| Line Search Steps | - | 10 | - | - |
| CG Steps | - | 100 | - | - |
| CG Damping | - | $10^{-2}$ | - | - |
| CG Tolerance | - | $10^{-10}$ | - | - |
| LR Schedule | - | - | Linear to 0 | - |
| Clip $\epsilon$ | - | - | 0.2 | - |
| PPO Epochs | - | - | 10 | - |
| Value Coef. | - | - | 0.5 | - |
| Train Freq. | - | - | - | 1 step |
| Gradient Steps | - | - | - | 1 |

Table 4: Hyperparameters used for the MuJoCo experiments.

In this section, we detail the experimental setup of the MuJoCo experiments, as well as adding new plots.

**Hyperparameters.** Table 4 lists the hyperparameters used for each Deep RL agent on the MuJoCo benchmark. Each agent is trained during $10^6$ interactions with its environment in the cases of HalfCheetah-v3, Hopper-v3, and Walker2d-v3. It is trained during $2.10^6$ interactions in the cases of Ant-v3 and Humanoid-v3. In all cases, a training episode is made of no more than $10^3$ interactions.

**Scores.** According to algorithm 3, after each agent is trained, the resulting policy performs 50 evaluations episodes. The score of the agent is the mean performance on these 50 episodes.

**Learning curves.** Fig. 11 presents sample efficiency curves for all algorithms in each environment. The shaded areas represent 95% bootstrapped confidence intervals, computed using `rliable` (Agarwal et al., 2021). Note that each curve may be an aggregation of a different number of scores. The number of scores can be found in the bottom right table of Fig. 10.

**Additional comparison plots.** Fig. 10 expands upon the comparisons given in the main text (in Fig. 5) by also plotting the score distributions of each agent using boxplots.

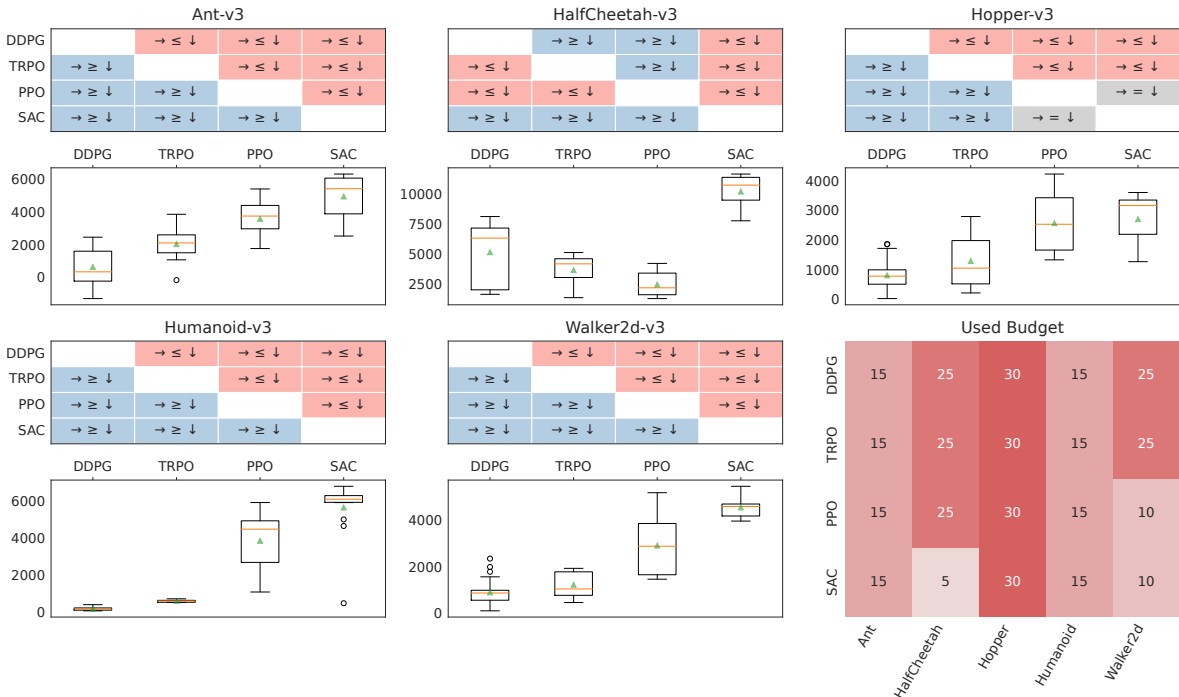

Figure 10: ADASTOP decision tables (top) and score distributions (bottom) for each MuJoCo environment, and the budget used to make these decisions (bottom right). The medians are represented by green triangles and the means by horizontal orange lines.

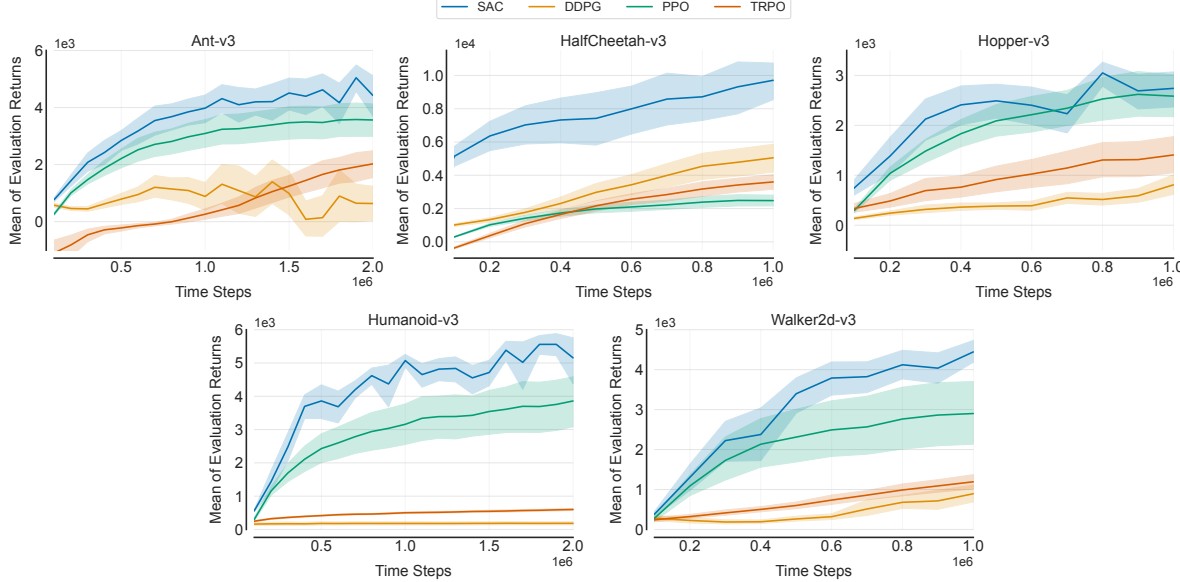

Figure 11: Mean of scores with 95% stratified bootstrap confidence intervals. Note that the curves in one figure may use a different number of scores, depending on when ADASTOP made the decisions.

## H.3   Additional plot for Section 5.2

To illustrate Section 5.2, Figure 12 represents the kernel density estimation of the distribution of scores from (Colas et al., 2019).

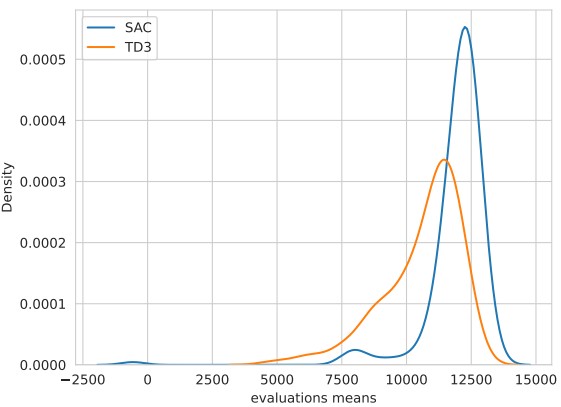

Figure 12: Distributions of scores for a SAC agent and a TD3 agent on HalfCheetah obtained with 192 independent scores. Each score is obtained on an agent that has been trained during $2.10^6$ interactions.

