# OpenReview forum: "AdaStop: adaptive statistical testing for sound comparisons of Deep RL agents"
_TMLR — Accepted by TMLR_

### Review · Reviewer_9doG · 2024-02-15

**Summary Of Contributions:**

This paper addresses the problem of comparing the performance of algorithm implementations with small sets of noisy performance samples. They develop the AdaStop algorithm that uses random permutation testing to iteratively check if the implementations under comparison can be separated or if more samples need to be gathered. AdaStop can also be used to process previously gathered data to perform a permutation testing analysis. This procedure is shown empirically to produce informative statistical comparisons between deep reinforcement learning algorithms with fewer samples than previously investigated statistical tests.

**Audience:**

Yes

**Broader Impact Concerns:**

No broader impact concerns.

**Claims And Evidence:**

Yes

**Requested Changes:**

It would strengthen the paper if awkward or confusing portions of the paper were improved.

How does the stochasticity of the score metric impact the confidence of agent comparisons? The definition of score at the end of Section 2.1 acknowledges that scores are typically the mean return of a policy across a set of episodes in practice, but this is merely an estimate of a policy's expected return. How do these errors interact with the confidence and power of the AdaStop analysis, and how do these errors interact with the error induced by sampling permutations? Are they entirely independent? I think this is an important discussion for the paper to include.

**Strengths And Weaknesses:**

The core idea of using random permutation testing combined with sequential group testing to compare algorithm implementations is logical and well motivated for the types of comparisons typically performed in the RL field. The example figures in the first part of the paper are effective illustrations for understanding aspects of AdaStop. The experimental results are convincing that AdaStop is more sample efficient than current approaches. These are straightforwardly strong aspects of the paper and they represent the majority of my overall opinion of this work.

However, I do have various critiques and questions.

1. Distinguishing between algorithms and implementations/agents in Section 2.1 is helpful, but a missing distinction is between agents and "trained agents". Perhaps a trained agent should be labeled a "policy"?
3. Regarding the conclusion
> Compared to the fixed budget setting, we allow a larger error rate, which results in a test that is more sample efficient than bandit algorithms

    at the end of Section 3.1, the wording makes it unclear to me if it is claiming that AdaStop is more sample efficient than all bandit algorithms or only those with a fixed budget. I would guess the latter, so is the trade-off then that fixed budget bandit algorithms will always exhaust their budget and thereby achieve lower error rates?

    How does AdaStop compare with fixed confidence bandit algorithms? Presumably, fixed confidence bandit algorithms could stop earlier than AdaStop since AdaStop must (actually, is this a strict requirement?) sample the same number of scores for each agent involved in active hypotheses, whereas bandit algorithms sample a score from one agent at a time. Does this restriction on AdaStop's early stopping ability have any statistical benefit?

4. The discussion around how many unique values of $T_{N, k}$ there are across all the permutations is good, but one sentence is awkward and somewhat ambiguous. By ". . . there are $\frac{1}{2} {2N \choose N}$ possible permutations in the first interim that give unique values to $T_{N, 1}$, I think you mean that there are $\frac{1}{2} {2N \choose N}$ unique values for $T_{N, 1}$ and there are not $\frac{1}{2} {2N \choose N}$ permutations that are special in some way from the permutations that yield the same values.

    Can you not use the fact that there are fewer unique values across the permutations than the number of permutations to get more accurate estimates of the test statistics for the same computation even when $\frac{1}{2} {2N \choose N}^k$ is large, *e.g.*, a stratified sampling approach where unique values are sampled and combined with appropriate weights instead of sampling permutations uniformly. Furthermore, have you looked into whether or not it is faster to sort the sampled permutations according to the number of alterations from the identity permutation and applying only necessary alterations to compute the value for each permutation rather than computing each from scratch?

5. There are various awkward or confusing wordings. Examples:
    1.  > The fact is that if we can be confident that 3 executions is not enough, how many executions would be enough in a statistical significant way?
    2. >  On the other way around, are 80 executions not too much to draw a statistically significant conclusion considering that each execution consumes resources (energy, CPU time, memory usage, etc) and contributes to pollute our planet and to the climate change.
    3. > Moreover, we want the conclusion of the comparison to be reproducible: if someone redoes the comparison of the same algorithms on the same task, the conclusions should be the same: this concept is known as statistical reproducibility . . .
    - line 4 of Algorithm 1, "running $N$ times the trained agent $A_l$" -> "running trained agent $A_l$ $N$ times"


6. Other sources of confusion:
    1. Line 8 of Algorithm 1, why is it not simply `return reject`? And then, if both the reject and accept conditions are within the main loop, are the conditions after Line 14 still required? It looks to me like they can be removed.
    2. The definitions of $\mathcal{S}_k$ and $\hat{\mathcal{S}}_k$ implicitly depend on $N$, which is a variable in the experiments. Why not annotate them with $N$?
    3. In Table 1, why is the power much larger given a fixed number of scores when $N$ is small? Specifically, $(N=2, K=2)$ < $(N=1, K=4)$, $(N=2, K=3) < (N=3, K=2) < (N=1, K=6)$. Except for $(N=3, K=2)$, which occasionally computes fewer scores because of early stopping, each of these tests have the same amount of data to work with on their last $k=K$ iterations. I would expect them to be the same on average.

        Related, if $(N=3, K=2)$, occasionally computes fewer scores because of early stopping, why does $(N=1, K=6)$ appear to never stop early? Should not $(N=1, K=6)$ be more likely to stop early?
    4. The comparison pairs, $(c_i)_{i \le L}$, defined as parameters to Algorithm 3 are unused. Related, $J$ in Line 1 of Algorithm 3 appears to be undefined.
    5. Line 4 of Algorithm 3 generates $N$ trained agents/policies for a given agent and then Line 5 states "Collect the $N$ scores of agent $A_l$". What if the policies or the environment are stochastic?

---

> ### Author Response · Authors · 2024-02-27
> **Answer to Reviewer 9doG**
>
> Hello,
>
> First, we want to thank the reviewer for his careful reading and comments.
>
> **Answer to questions:**
>
> 1. We thank the reviewer for this suggestion. We will introduce the use of the term "policy" defined as a "trained agent" in the revised version of the paper. This has been helpful in particular when addressing stochastic policy as discussed hereafter.
> 2.
>  - Regarding the comparison with fixed-budget bandit algorithms. Yes, it is as you write: fixed budget bandit algorithms will always exhaust their budget and thereby achieve lower error rates. We will rephrase to make it clearer.
>  - About fixed confidence bandit algorithms: regarding the theoretical guarantees, fixed confidence algorithms have typically better theoretical guarantees although in practice permutation tests are better for small sample sizes (typically we use a maximum of 30 scores while bandits algorithms typically deal with thousands of samples). The requirement of sampling the same number of scores could be relaxed. Indeed, it would make sense to use interims containing more scores for an algorithm that runs faster (i.e. use $N=5$ for SAC which is costly and $N=10$ for PPO). For now this must be chosen in advance as AdaStop is not a bandit algorithm, having adaptive size of batches is a future work.
> 3. You are right: we reformulated to "we enumerate that there are $\frac{1}{2} {2N \choose N}$ possible unique values to $T_{N,1}$" (p11). Remark that we could use a restricted number of permutations, even without stratification. The type I error proof would still hold, although the power could suffer if the choice of permutations was not made properly. The stratification approach is a good solution to keep a powerful test while reducing the number of permutations. We did not implement it because in practice AdaStop already worked well with a reasonable computational time with a simple Monte-Carlo approach.
> 4.  We thank the reviewer for catching these. We fixed the wording. More generally, we checked the whole paper.
> 5.
>    1.: You are right, we fixed this.
>
>    2: We now annotated both $\mathcal{S}$ and $\widehat{\mathcal{S}}$ with both $k$ and $N$.
>
>    3: We believe that this is related to the discrete nature of permutation tests. When $N$ and $K$ are both small, it can happen that AdaStop does not use the whole budget of $\alpha$ and this will imply a much lower power. For instance, if $N=2,K=2$ then at first interim, we have $\lfloor \frac{1}{2}{2N \choose N} \alpha/2 \rfloor = 0$ hence the first boundary is necessarily infinity and never rejects. Then at the second interim  $\lfloor \frac{1}{2}({2N \choose N})^2 \alpha/2 \rfloor=3$ so that in effect we only used a budget of $\alpha_{eff} = \lfloor \frac{1}{2}({2N \choose N})^2 \alpha/2 \rfloor / ( \frac{1}{2}({2N \choose N})^2)\simeq 0.041$. This is an edge case happening in the first few interims when $N$ is small, and it causes some strange readings in Table 1, we added a small discussion about this page 14.
>
>   4: There is a typo, it should be $c_j$ for $j\le J$ and not $c_l$ for $l\le L$. The $c_j$'s are the comparisons we want to make and that are used to define the test statistics, it is represented in the notations by the ${(j)}$ exponent in $T_{N,k}^{(j)}$. $J$ is defined in the case of AdaStop at the beginning of Section 4 and denotes the number of tests (or equivalently the number of comparisons) that AdaStop performs.
>
>   5: If the policy or the environment is stochastic, we still use the same process as we suppose that this is taken care of when the score is computed. Typically, the score is a mean of evaluations and by using a large number of evaluations the stochasticity of the policy or the environment has less impact on the score. Please note that this does not affect the theoretical analysis as the test is still valid even if we used only one evaluation as a score. AdaStop does not care where the stochasticity originates from. AdaStop only relies on the fact that stochasticity is independent among the scores.
>
> **Requested changes**
>
> We agree that the impact of stochasticity of score metric is an important question practically speaking. As explained before, the theory would not change if we used very few evaluations for each score but on the other hand, the power of the test would suffer a lot. In practice, it is often the case that evaluation is a lot cheaper than training, and it is acceptable to request for 20 to 50 evaluations for one score depending on the available computational power. We think that choosing sufficiently large number of evaluations (e.g. 50 evaluations) should answer this in practice, even though we agree that this is not entirely satisfactory. Addressing quantitatively the effect of stochasticity of score metric is kept for future work. We have added a remark on this just above Section 2.2.
>
> We hope this answers your questions.
>
> Best regards. The authors of paper #2061.

---

### Review · Reviewer_cbp8 · 2024-02-17

**Summary Of Contributions:**

This paper addresses the problem of comparing empirical results in deep RL (rather the procedure is more general but presented for the deep RL setting) to state, with some statistical confidence, whether an algorithm performs better than another. It proposes AdaStop, a sequential testing procedure that works for simultaneous comparisons of multiple algorithms and automatically stops the collection of new runs whenever a desired confidence on the test is reached. The paper provides both theoretical guarantees on the family-wise error of the tests and an empirical evaluation in toy domains and Mujoco tasks. Finally, the paper provides an implementation of AdaStop that aims to become a standard statistically-grounded procedure for empirical analysis of deep RL algorithms.

**Audience:**

Yes

**Broader Impact Concerns:**

This paper categorises as fundamental research, for which a specific statement on potential societal impact does not seem necessary.

**Claims And Evidence:**

Yes

**Requested Changes:**

I do not have specific changes to request (perhaps aside from the minor reported above) but rather clarifications on my comments (especially in the Weaknesses section).

**Strengths And Weaknesses:**

Strengths
- The paper is very well-written and remarkably easy to read. All of the building blocks are presented with clarity and details. Indeed, the paper may be even too verbose in the presentation of the background (Section 3) and looks like the text could have been streamlined into a "short" submission (less than 12 pages).
- The paper tackles an old and extremely important problem in deep RL, i.e. how to compare empirical results with statistical confidence. Its contribution is mostly orthogonal to (Agrawal et al. "Deep RL at the edge of the statistical precipice"), which focuses on performance metric rather than number of independent runs.
- Differently from most previous works, it aims to provide a running solution (coming with code) rather than serving as a warning for bad practices.

Weaknesses
- Novelty: This may not be an actual weakness, but I think the paper is not doing a great job in highlighting what is a novel contribution. From my understanding AdaStop combines permutation testing and multiple sequential tests, but it is unclear what is novel beyond the combination itself. Also, are the theoretical results novel or directly derived from previously known guarantees?
- Is the guarantee of AdaStop tight in any sense? It would be extremely valuable to get a sense (e.g., a lower bound on the number of runs required to get statistical significance) of how far is from optimum, as sub-optimality entails the experimental analysis of paper implementing AdaStop will incur in severe and unnecessary costs of computation and time. The empirical evaluation shows between 15 and 30 runs for simple Mujoco tasks, from which it is fair to wonder whether statistical significance in deep RL is out-of-reach for academic institutions without almost unlimited computing power.
- From my understanding, the paper provides only an asymptotic (in the number of runs N) theoretical guarantees on the family-wise error for mean comparisons with AdaStop. This would not be ideal, as most deep RL works are concerned with comparing the mean score rather than the distribution.
- Another potential limitation of the procedure and analysis, is that while AdaStop is adaptive on the number of runs, it does not adapt to the number of training samples. However, this seems to be crucial in RL, where a difference between algorithms performance can arise at later stage of training.

Minor
- The summation in the definition of B_N includes a space of permutation (?) that has not been defined?
- In Table 1 the shaded bars go out of the table.
- Captions of the figures in Section 5 are not really detailed, which means the reader has to rely a lot on the text to understand what is shown there.

---

> ### Author Response · Authors · 2024-02-27
> **Answer to Reviewer cbp8**
>
> Hello,
>
> First, we want to thank the reviewer for his careful reading and comments.
>
> **About the weaknesses:**
>
> - Novelty: The theoretical results are new up to our knowledge. Moreover, they are uncommon in that they consider a sequential and nonparametric setting which is not often seen in the literature. To the best of our knowledge, these are the only results matching exactly the setting we address (multiple testing, sequential with finite budget and controlled family wise error, fully nonparametric).
> - The guarantees of AdaStop are only given on the Family-wise error. Hence, it is not clear for now how large is the power of the test,  this is in fact a work in progress for us, and we prefer to keep this as future work because this is very technical, there has been recent work (notably Minimax optimality of permutation tests, Kim et. al 2022) proving bound on the power for permutation tests and proving that they are efficient non-asymptotically, but these results are for non-sequential tests.
> - About the necessity to use 15 to 30 runs for Mujoco tasks, this was already highlighted in previous works on the subject (see in particular Colas et al. 2018). Although this may seem problematic for academic work we feel that this requirement should be more well known in the community, and this could be accommodated with a careful design of experiments in Deep RL. In particular, this would allow researchers to make an informed choice for benchmark environment on which we do the experiment (e.g. atari vs mujoco vs classic control) and choice of maximum budget. The final goal here is to have an experiment design that is at the same time practically doable with limited academic resources with strong statistical guarantees about the conclusions.
> - We discuss the limitations due to permutation test in testing the mean in page 12. We believe that Theorem 2 that states the asymptotic efficiency of the comparison of the means could be made non-asymptotic under concentration hypotheses (typically sub-Gaussian). We keep this for future work as this is rather technical in particular due to the theoretical complexity (permutation test, group sequential...) of AdaStop.
> - About the adaptation to the number of training samples, please note that AdaStop cannot be used for such a task because AdaStop only deals with independent scores. This is the most important requirement of Adastop, and it is not verified when trying to stop early during training. A lead to deal with this early stopping in training is to test for convergence of the training phase and use AdaStop only on a policy that converged (provided the policy converges).
>
> **Minor comments**
>
> We fixed these typos and problems in layout, thank you for catching these. We will add more details to the captions.
>
> We hope this answers your questions.
>
> Best regards. The authors of paper #2061.

---

> > ### Comment · Reviewer_cbp8 · 2024-02-29
> >
> > I want to thank the authors for their replies and for accommodating suggestions in the paper.
> >
> > Just a quick follow-up on novelty, I want to rephrase the question: Are the theoretical results a straightforward combination of previous derivations on permutation tests and multiple sequential tests? If not, what are the additional challenges that has been overcome and how?

---

> > > ### Author Response · Authors · 2024-02-29
> > > **Answer about the novelty of theoretical results**
> > >
> > > Hello,
> > >
> > > The theoretical results are indeed a combination of previous derivations on permutation tests, multiple testing and group sequential tests. However, we do not think it is a straightforward application of previous results. In particular, in proving Theorem 1, the main difficulty was to handle the various dependencies (due to sequential treatment and due to multiple testing) properly. Due to these dependencies, we needed to investigate properly the joint law of scores (Lemma 2) and make all the verifications to be sure that the permutation argument goes through. To prove Theorem 2, we similarly needed to control uniformly and jointly the convergence of the test statistics (Lemma 4). These were the principal difficulties in proving these two theorems.
> > >
> > > Best,
> > > The authors of paper #2061.

---

### Review · Reviewer_qLrz · 2024-02-22

**Summary Of Contributions:**

The paper proposes AdaStop, an evaluation protocol which uses a nonparametric statistical test to efficiently compare RL algorithms' performances. AdaStop is based on a permutation test, and can be applied to compare multiple algorithms simultaneously.  The paper bounds the probability that the method will falsely state that the performance distributions induced by two algorithms are different, and shows that the test asymptotically will also identify the equality of distribution means to a specified significance level. The paper evaluates AdaStop on four RL algorithms on the MuJoCo benchmark, along with synthetic experiments comparing pre-specified distributions.

**Audience:**

Yes

**Claims And Evidence:**

No

**Requested Changes:**

My biggest concern with the paper is that the proposed test does not actually answer the question of whether one algorithm is better than another, but instead answers the question of whether one algorithm induces a different final performance distribution than another. Given that the distributions over returns induced by deep RL algorithms are often highly variable, I worry that the proposed methodology might inadvertently lead to many researchers falsely claiming that they have 'statistically significant' results showing that their method outperforms a baseline when in fact their method simply produced a different distribution over performances that happened to have a slightly higher sample mean.  The following would help to address this concern:

1) An empirical evaluation of the robustness of AdaStop to distributions which have similar means but very different `shapes' (e.g. mixture of Gaussians with different numbers of modes and mode locations).
2) Principled guidance on how to interpret the results of AdaStop if the algorithm terminates quickly but the sample means are very close.
3) Non-asymptotic guarantees on the 'false positive' rate of the algorithm (potentially under certain easy-to-verify conditions on the distributions being tested), or at least a broader evaluation on more deep RL algorithm/environment combinations which includes a comparison between the mean performance gap between two algorithms and the number of samples requested by AdaStop (in particular, it would be encouraging to see a strong negative correlation between these two quantities).


The following points would be nice to have but are less critical to my evaluation.

- It would be useful to elaborate on the choice of $\hat{S}_k$ in equation 2 when it is introduced -- when I reached equation 2 I had to stop reading and prove the correctness of the formula to myself before continuing, which broke the flow of the paper. Even just describing the mutually exclusive events whose probabilities you are computing implicitly in the equation would help to give a reader unfamiliar with permutation tests such as myself a better intuition for Algorithm 1.
- One point in AdaStop's favour is that when evaluating a deep RL algorithm against a set of baseline algorithms we typically only care about whether the new algorithm beats the baselines, i.e. we do not care about the relative ranking of the baselines. This means we go from having $O(N^2)$ hypotheses (is algorithm i better than algorithm j for all i,j pairs) to falsify to having $O(N)$ hypotheses (is algorithm $j$ better than algorithm 1 for all $2 \le j \le N$), which would consequently reduce the number of algorithm evaluations needed.
- Would it be possible to adaptively select the 'batch size' used to generate new data? In a domain like the Arcade Learning Environment, running a single experiment can take several days, and so if it is likely that more than one batch of data will be needed to falsify a hypothesis to a target significance level, it would be useful to run many iterations in parallel.

**Strengths And Weaknesses:**

**Strengths:**

- The paper concerns an important topic, as we currently don't have a rigorous methodology as a field for identifying whether one algorithm is better than another

- The proposed test is non-parametric, and so doesn't depend on hard-to-satisfy assumptions on the properties of the score distribution. This is especially useful for a setting like RL where score distributions for an algorithm are often multimodal / very different from standard e.g. Gaussian distributions.

- The proofs appear to be correct and are easy to read.

- The paper is well-written, providing suitable background on the relevant hypothesis-testing literature to understand the proposed method, and a clear explanation of the AdaStop algorithm.

- Although the number of permutations needed to evaluate the AdaStop algorithm is quite large even in the approximate setting, given the relative cost of evaluating a deep RL algorithm this does not seem like a particularly meaningful computational burden.

- The paper provides theoretical guarantees on the method which bound the probability of rejecting a correct hypothesis, and which suggest that in the large-sample limit the test is also able to evaluate the equality of distribution means with the target level $\alpha$.

 - The method is friendly to batching, which is useful for the deep RL settings where a single experiment iteration can take a great deal of time.

- The empirical evaluations on the MuJoCo environment show that AdaStop is able to calibrate its sampling procedure to each environment.


**Weaknesses:**

- The paper abstract claims that AdaStop can answer the question 'is my algorithm the new state of the art' but this isn't exactly true -- know that deep RL agents are often trained on multiple environments, and AdaStop doesn't answer the question of whether the aggregate performance is the new SOTA. E.g. in Atari it is very common for an algorithmic improvement to improve performance on most but not all games.

  - Also, in the low-sample regime this method will not answer the question of whether the algorithm has a higher mean than the state of the art, but rather whether the algorithm comes from a different distribution (see point 3 below for a more detailed elaboration on this concern).

- The proposed family of statistical tests does not seem very amenable to power analysis -- this seems like a problem, since if I run this test on an algorithm and it returns saying that the data does not support rejecting the null hypothesis with the desired confidence level, then I don't know how to update my belief that the algorithms really do have the same mean.

- The proposed statistical test gives an answer to the question of whether two distributions are the same or not. However, for some distribution pairs it is easy to tell that they are not the same while also difficult to tell which has a greater mean. It is possible, for example, that if my new algorithm has much larger variance than a baseline but a very similar mean performance, a permutation test will identify that the two algorithms produce different return distributions with fewer samples than would be necessary to disentangle the means. As a result, I might (over-)confidently conclude that because AdaStop says to reject the null hypothesis and the sample mean for my method is higher than for the baseline, my algorithm has been robustly demonstrated to outperform the baseline.

---

> ### Author Response · Authors · 2024-02-27
> **Answer to Reviewer qLrz**
>
> Hello,
>
> First, we want to thank the reviewer for his careful reading and comments.
>
> **About the weaknesses:**
>
> *Difference between comparing distributions and comparing means.* These limitations are discussed in the article (see end of Section 2.3 and discussion "Hypotheses of the test" in Section 4.1). We detail the point about the power later in answer to point 3.
> *About the power analysis,* we are currently working on it and we expect to submit the analysis very soon. AdaStop uses the empirical means and as such the test is indeed approximately about the mean, and with sufficient concentration one can prove a type I and a type II error bound in a non-asymptotic regime. The proof techniques already exist when one does not look for a tight bound. In that case, one can use the two moments method (See Section 4 of "Minimax optimality of permutation tests" (Kim et al. 2022) for its application in the non-sequential case). The difficulty comes when one wants to derive a tight bound with exponential speed because such an analysis is based on isoperimetric inequalities for random permutation and this is a bit technical (See Section 6 of "Minimax optimality of permutation tests" (Kim et al. 2022), but we are confident that we will manage to use such techniques for a power analysis of AdaStop.
> *About an overconfident rejection decision.* For now, the theory of AdaStop just says that if the distributions are the same, then there is a high probability that we do not reject the null hypothesis. It does not say anything about the case where the distributions are different. In practice, in the toy examples of the submitted paper, we see that AdaStop is not very confident in rejecting the null hypothesis when the means are the same but the distributions are different (see Figure 4). We believe in fact that what the reviewer describes cannot happen as soon as there is sufficient concentration of the distribution around their mean (typically, one can think of finite variance hypothesis), but this is not explicit in the theory for now due to the asymmetry of statistical tests.
>
> **Requested changes**
>
> 1. We believe that an empirical evaluation of the robustness of AdaStop to distributions which have similar means but very different `shapes' is already provided in Figure 4 of the paper in which we applied Adastop to various mixtures of Gaussian distributions with the same mean but different distributions. This experiment is extended further in Figure 7 (Section H.1).
> 2. If AdaStop stops early, the means should not be very close to each other provided that the distributions concentrate well around their means as already explained in the discussion about the power of AdaStop. An example of this can be found in Figure 9 in the appendix in which either the means are "close" with small std in which case AdaStop can conclude that there is a difference. On the other hand, in the case of Hopper environment in Figure 9, PPO and SAC have close means and hence the decision is to accept $H_0$.
> 3. Non-asymptotic power guarantees have already been discussed in the above discussion about the power of AdaStop.
>
> About the other points:
>
> - We added an explanation about the computation of the boundary $B_{N,k}$ in equation (2).
> - The reviewer is right that in the case of comparison with baselines agents, if we do not care about the ranking of the baselines agents, we can drastically reduce the number of tests to perform. We added a remark on this in the paper (see p9, below Definition 1). On the other hand, it may be interesting to do all the pairwise comparisons in the case of testing the effect of the change of a parameter, for example, to study the effect of the discount factor on an algorithm.
> - The batch size $N$ has to be chosen in advance in AdaStop. It cannot depend on the data, but it can be chosen appropriately large in order to use all the available computational. Remark that this is a limitation compared to batch bandits approaches as adaptivity of batch size is one of the ingredients needed to reach optimality. This however simplifies the algorithm and does not make a big difference when the size of the batches is small.
>
> We hope this answers your questions.
>
> Best regards. The authors of paper #2061.

---

> ### Comment · Reviewer_qLrz · 2024-03-02
> **Clarifying concerns about permutation tests**
>
> Thanks to the authors for the quick response. I appreciate the updates to the paper and will take a more detailed look over the next couple of days. However, I should emphasize that my concern over the validity of using a test on distributional equality to compare means did take into account the discussion in the paper and Figure 4. Although these do provide some data points supporting the use of this test, I don't think they provide a sufficiently broad evaluation of the types of distributions one might want to compare in practice.
>
> To make my concern clearer, consider the following example: we have one distribution P1 which is a standard normal. We have a second distribution P2 which assigns 90% probability to the outcome -1, and 10% probability to the outcome +9. These two distributions have the same mean (0), but if I take 5 samples of each there is a fairly high probability that the empirical distribution of P2 will not contain the extreme positive outcome. This is roughly analogous to the setting where I have one RL algorithm which sometimes attains extremely high performance but usually fails to take off, versus a second which consistently gets mediocre performance. Even though both distributions have the same mean, with fairly high probability the samples gathered from distribution P2 will lead the permutation test to incorrectly reject the null hypothesis after the first round of N=5 samples even for low levels $\alpha$. When I ran this setup empirically for an even more extreme variant (-10 with probability 0.9 and +90 with probability 0.1), I ended up with samples where gap between sample means for the two distributions was greater than _all_ of the sample means for the random permutations in ~60% of the random draws, meaning that the test would reject the null hypothesis for any $\alpha > 0$.
>
> Figure 4 does show that as the distance between modes of the mixture distribution increases, the probability of falsely rejecting the null hypothesis also increases, but it stops at a relatively small gap in the distribution means. What the example I provide above shows is that when the distribution is both high-variance and `lopsided', it is possible to construct comparison pairs where, for the 5-sample iterative protocol used in the paper, the permutation test rejects the null hypothesis after the first iteration most of the time. It is possible to debate how realistic this type of distribution is, but since RL algorithm performance often does exhibit this type of 'all or nothing' behaviour, particularly if the algorithms have different exploration strategies, I think it is reasonable to expect the paper to discuss this type of failure mode.

---

> > ### Author Response · Authors · 2024-03-05
> > **Answer to concerns about permutation tests**
> >
> > We thank the reviewer for this discussion, in response, we have conducted an additional experiment and incorporated it into Figure 4. This experiment was inspired by the reviewer's suggestion. The specifics of this experiment are detailed in Case 3 of Section 5.1 and illustrated in Figure 4(c).
> >
> > In Figure 4 (c), we see that the permutation performs well when the gap between the two modes is not too large. If on the other hand, the gap between the two modes is very large as you suggest, we think that it would be meaningful to use robust mean estimation in order to actively ignore the rarely seen extreme cases, this is what was proposed in the article  [(Agrawal et al. 2022)](https://arxiv.org/abs/2108.13264). Using robust mean estimators is possible and in fact easy to do with permutation tests, see [(Chung and Romano, 2013)](https://arxiv.org/abs/1304.5939), although we prefer not to include it in the article for simplicity.
> >
> > Best, The authors of paper #2061.

---

> ### Comment · Reviewer_9doG · 2024-03-18
> **Clarifying decision tables and related concerns about distribution testing**
>
> Thanks for the responses.
>
> Could you please clarify how the decisions in the decision tables are being constructed? I am having trouble finding it now. I may be misremembering but my current thought is that when A is judged to be different from B, you compare the average score of A to B to arrive at the decision. Is that correct?
>
> If so, I am afraid that AdaStop is not providing statistical confidence on the decision table results. For example, if AdaStop rejects the hypothesis that SAC generates the same scores as PPO and the average score of SAC happens to be higher than PPO on these particular samples, can we really have confidence that SAC is better on average in this domain? It seems to me now that all of the confidence could stem from the fact that the distribution of scores from SAC and PPO are different, in which case the difference in their sample means may be misleading. Not even in that their means are the same, but that they are actually reversed from the sample estimate.
>
> Of course, the more sample scores, the more accurate the sample mean comparison will be, but the idea is that permutation testing could allow us to generate fewer sample scores. In fact, is it not possible that AdaStop's early stopping could prevent us from making accurate comparisons? That is, AdaStop could confidently detect early on that SAC, for example, has a very different score distribution than all the other algorithms and thus stop running SAC instances. However, once we go to compare mean scores to actually come up with ordering decisions, we end with very low confidence because we have very few SAC scores?

---

> > ### Author Response · Authors · 2024-03-20
> > **Answer to concerns on distribution testing**
> >
> > Hello,
> >
> > For the construction of the tables and for AdaStop algorithm, we always compare the empirical mean. As the reviewer says, the theory we present does not give statistical significance to the test between the theoretical mean of the distributions when the sample size is small (which is our case) but as it is common to do in nonparametric tests, we use a test on the distributions as an approximate test on the mean. We are currently working on having theoretical results for the comparison between means, but we keep this for future works.
> > We want to insist however that in addition to the approximation theorem which may be weak because it is asymptotic, AdaStop has been shown to be empirically efficient.
> >
> > To re-use the example of the reviewer, if we are testing SAC against PPO and the average score of SAC happens to be higher than PPO on this particular sample, this is possible. This is the type I error and this happens approximatively 5% of the times (see Figure 4). The 5% encompasses the error that the algorithm makes whatever the cause of the error (wrong early stopping and/or wrong rejection). As always in a statistical test we are never completely confident in our decision, there is an intrinsic error that cannot be avoided, an event of small probability that give a wrong decision.
> >
> > Moreover, we also want to highlight that the test is designed to stop and reject early only when the empirical means are very different, and with high probability this should not happen as soon as there is sufficient concentration of the distribution around its mean. If there is no sufficient concentration of the distribution around its mean, then it is anyway very difficult to test the equality of the means whatever method is used.
> >
> > Best, The authors of paper #2061.

---

### Author Response · Authors · 2024-02-27
**General answer to reviewers**

Hello,

We want to thank the three reviewers for their careful reading and fruitful remarks on our article that allowed us to improve the overall quality of the article. We modified the paper to reflect the reviewers comments and uploaded the revision on openreview, we also uploaded in the supplementary materials a diff pdf highlighting the changes between this and the initial version of the paper.

As a general comment we want to point out that the aim of this article is to be used by practitioners and as such we did not push the theory as much as we could have.
Instead, we decided to keep the more theoretical questions for later work. We are currently writing an article about the non-asymptotic guarantees regarding the power of AdaStop, this article will be submitted separately as a theory paper.

Below, we answer each reviewer separately.
Best regards,
The authors of paper #2061.

---

### Decision · Action_Editor_UamV · 2024-05-06

**Recommendation:** Accept with minor revision

**Comment:**

The reviewers overall liked the paper, and appreciate the utility in focusing on more rigorous experimentation in RL. There is, however, a concern around the fact that many readers might miss the nuance that there is a comparison of distributions rather than means. There was a lengthy discussion, and we decided on Accept, but under the conditions that this potential issue is addressed.
1. The authors more clearly state upfront that distributions are compared, rather than means, and the ramifications of this choice.
2. It is better to tone down claims stating that AdaStop can allow practitioners to determine if their algorithm is the new state-of-the-art with statistical guarantees

It is useful here to include snippets/paraphrasings of our discussion.
From a reviewer
"After the discussion with the authors and the other reviews, and after re-reading the paper, I have decided to recommend rejecting this paper. I am concerned that this paper will be misused by practitioners to make claims about statistical significance and confidence that are inaccurate or misleading.
...
Strictly speaking, AdaStop can only verify a common assumption: that different algorithms produce different score distributions. Except under the strong assumptions of Theorem 2, all statistical claims when using AdaStop must be phrased in terms of score distributions. As reviewer qLrz correctly pointed out, there are common practical scenarios where knowing that score distributions differ tells you nothing about expected scores. Since the paper claims that AdaStop is a solution to identify algorithms that perform better in a statistically significant way without assumptions on the algorithm score distributions, it is a major problem that achieving statistical confidence on such decisions do actually require score distributions to be sufficiently concentrated around their means. In fact, this concentration assumption is not merely that most of the probability mass is near the mean (as it is in Case 3 of Figure 4), but that there is no substantial mass far from the mean.

The authors say on page 12 that "[doing a test on distributions and ranking based on sample means] is justified because without strong concentration assumptions on the distributions, we lack control on the mean of the distributions." I have two responses to this claim. First, lacking the means to make a statistically meaningful statement about the difference of means does not justify the claim that a small sample test on distributions tells us anything about the difference in means. Second, bootstrap sampling can actually construct confidence intervals and tests for differences in means without making any assumptions on the score distributions.

Why would we use AdaStop over bootstrap sampling? The paper's answer on page 15 is that AdaStop makes a decision using fewer scores. In the example, they say that AdaStop uses 12 instead of 15 scores to achieve an empirical power of 0.82. They compare their empirical power to the power of Welch and boostrapping tests, but it is unclear to me if they really have the same meaning. I do not believe we have any guarantee about the accuracy of AdaStop's empirical power. Perhaps a more direct comparison of the accuracy of AdaStop's empirical power with bootrapping's power could have been useful here.

Overall, I find that this paper over-claims that AdaStop can allow practitioners to determine if their algorithm is the new state-of-the-art with statistical guarantees by failing to adequately address the issue that the guarantees AdaStop provides are not directly related to expected performance. This issue is subtle enough that I did not notice it on my first read through, and important enough that I recommend rejection to avoid future similar misunderstandings.

To give the paper its due, there is definitely something interesting about using permutation testing to compare algorithms. Ultimately, AdaStop is using the number of scores to complete a permutation test as an early stopping heuristic. I feel like there is still potential in using this heuristic, particularly combined with bootstrap sampling to construct confidence intervals, and completing more runs only where strong confidence is required. I look forward to the theory article on AdaStop, perhaps it will resolve the issues I have brought up here and reduce the potential I see for AdaStop to be misused."

Another issue raised by an author is the it is potentially implied that AdaStop is tight, and if 30 runs are required for AdaStop, then one could not get away with less. But, of course, a different statistical test might be able to. If there are any such claims, then do consider modifying them.

In summary, the key things to address are the two points I bulleted above. These are strictly writing changes, so an Accept with Minor Revisions is appropriate. I will check the changes. If you need to clarify the revisions before doing them, then feel free to post a question to me (the action editor).

**Audience:**

There is a clear audience for this work.

**Claims And Evidence:**

The claims are reasonably well-supported, albeit that there is a bit of overclaiming that needs to be tempered (discussed more below).

---

> ### Author Response · Authors · 2024-05-21
> **Minor revisions**
>
> Hello,
>
> We changed the paper in accordance with the revisions suggested by the reviewers and the AC. We thank the reviewers and the AC for helping us to make clear the main messages of our article. In particular, we added a section (Section 2.5: Limitations of AdaStop) that details early-on that we are comparing the distributions and not directly the means and also highlight that we do not claim to be optimal. We also added some explanations on the asymptotic results.
>
> Concerning the comparison with bootstrap approaches and the other methods we compare ourselves to, we want to emphasize that concerning the theoretical guarantees for such nonparametric approaches, most theoretical guarantees are, to our knowledge, only asymptotic. The classical bootstrap results are asymptotic and very similar in spirit to our Theorem 2 and the analysis from Section E.
>
> Best regards,
> The authors of paper #2061.

---

> > ### Author Response · Authors · 2024-06-06
> > **Camera Ready version**
> >
> > Hello,
> >
> > We de-anonymized the article and uploaded the camera-ready version.
> >
> > Best regards, The authors of paper #2061.